# CryoACE: An Atom-centric Framework for Accurate and Automated Model Building in Cryo-EM

**Minzhang Li** [1 2]  **Mingrui Li** [1]  **Weichen Qin** [1]  **Qihe Chen** [1 2]  **Sixian Shen** [1]  **Yuan Pei** [1]  **Jiakai Zhang** [1 2]  **Jingyi Yu** [1]

## Abstract

Protein automodeling from cryo-EM density maps faces unique challenges in enforcing physicochemical validity and managing conformational heterogeneity. Current solvers are often limited to static predictions or require computationally intensive heuristic searches. We present CryoACE, an end-to-end framework that reconstructs precise atomic graphs for both homogeneous and heterogeneous structures. Our method features two key innovations: an atom-centric reconstruction paradigm, where density features are sampled directly at atomic coordinates and iteratively recycled to refine structures—replacing expensive voxel convolutions for efficient multimodal fusion—and a training-free guidance mechanism that leverages predicted local resolution priors to resolve dynamic ambiguity. Validated on a newly constructed high-quality dataset, CryoACE significantly outperforms existing baselines on static benchmarks and, for the first time, unveils atomic-level dynamic conformations on complex real-world datasets like EMPIAR-10345 without relying on pre-built static structures. We release our code, model weights, and dataset to facilitate future research.

## 1. Introduction

Human perception of the three-dimensional world begins with raw observation but thrives on geometric abstraction. Modern vision foundation models illustrate this abstraction, with SAM (Kirillov et al., 2023) distilling visual inputs into precise masks and Grounding DINO (Liu et al., 2025) converting observations into structured bounding boxes. Beyond natural images, this capability empowers scientific discovery, particularly protein automodeling in structural biology. Here, the goal is to build precise atomic graphs directly from 3D experimental density maps (e.g., from X-Ray and cryo-EM). Resolving these volumetric signals into explicit atomic coordinates bridges the gap between physical observation and biochemical mechanism, serving as the cornerstone for structure-based drug discovery.

However, automated model building presents several unique challenges. First, biomolecule structures are subject to strict geometric and stereochemical constraints (e.g., bond lengths and angles) and must be consistent with their corresponding 1D amino acid sequences, requiring models to jointly reason over sequence priors and 3D observations. Second, experimental density maps are often noisy and highly redundant, making it computationally challenging to distinguish meaningful structural signals from background artifacts in high-dimensional volumes. Third, intrinsic conformational heterogeneity leads to averaged results by traditional methods (Punjani et al., 2017) or non-uniform resolutions within cryo-EM heterogeneous reconstructions (Zhong et al., 2021). Consequently, an effective framework must adaptively refine atomic structures according to local map quality and density variability, balancing structural accuracy with data fidelity across heterogeneous regions.

To tackle these challenges, traditional approaches like Phenix rely on iterative heuristic search, which is computationally intensive and sensitive to local minima, particularly with low-quality maps. Recent neural methods formulate this task from a 3D vision perspective, but often struggle to maintain geometric and stereochemical consistency. For instance, ModelAngelo (Jamali et al., 2024) may produce fragmented residue assignments, while E3-CryoFold (Wang et al., 2025) requires additional post-processing to refine geometry. Moreover, most existing learning-based solvers are primarily designed for homogeneous, single-structure reconstructions. CryoBoltz (Raghu et al., 2025) takes an important step toward modeling heterogeneity via training-free guidance. However, its backbone operates on pre-built structural representations rather than directly conditioning generation on density observations, which limits robustness

[1]School of Information Science and Technology, ShanghaiTech University, Shanghai, China [2]Cellverse Co, Ltd., Shanghai, China. Correspondence to: Jiakai Zhang <zhangjk@shanghaitech.edu.cn>, Jingyi Yu <yujingyi@shanghaitech.edu.cn>.

*Proceedings of the 43rd International Conference on Machine Learning*, Seoul, South Korea. PMLR 306, 2026. Copyright 2026 by the author(s).

under noisy or ambiguous maps.

Unlike prior pipelines that bolt density guidance onto sequence-only foundation models at inference time, our work introduces an atom-centric multi-modal architecture in which density features are sampled directly at predicted atomic coordinates and learned jointly with structure during training, establishing a physically grounded atom–density correspondence rather than a post-hoc refinement signal. We propose CryoACE, an **A**tom-**CE**ntric framework capable of generating physically plausible atomic structures in both homogeneous and heterogeneous scenarios of cryo-EM. To achieve this, CryoACE formulates model building as atom-centric coordinate generation, where density features are sampled at atomic locations and iteratively recycled to refine structures, aligning volumetric observations with atomic representations while avoiding expensive voxel-wise processing. Specifically, for multimodal feature extraction, we build upon the Boltz (Wohlwend et al., 2025) architecture by introducing an atom-centric coarse-to-fine density sampling strategy. Instead of brute-force voxel convolutions, we employ an atom-centric fine-grained sampler that leverages data recycling to extract high-resolution features around predicted atomic coordinates, improving both efficiency and precision. To handle conformational heterogeneity, we propose a novel training-free energy guidance mechanism including the global guidance and Q-guidance, yielding highly controllable predictions that faithfully align with raw cryo-EM maps. Furthermore, we integrate the direct prediction of atomic Q-scores and local resolutions into the framework. The predicted Q-score significantly accelerates inference, while the estimated local resolution serves as a critical prior for modeling heterogeneous reconstructions, enabling high-level control over the trade-off between modeling precision and density map guidance.

To train our model, we constructed a comprehensive, high-quality multimodal dataset comprising 10,915 entries, containing density maps, sequences, and half-map local resolutions. On homogeneous benchmarks, our extensive experiments demonstrate that CryoACE achieves state-of-the-art performance, significantly surpassing existing baselines on Q-score and RMSD. Moving beyond static tasks, we validate the effectiveness of our predicted local resolution maps. By leveraging these predicted local resolutions, which are often inaccessible via traditional means in heterogeneous settings, we tackle the challenging heterogeneous datasets EMPIAR-10345 and EMPIAR-10516. Our approach demonstrates superior recovery of dynamic atomic ensembles compared to current methods, unveiling the atomic-level dynamic conformations of these complexes.

## 2. Related Work

### 2.1. Conventional Methods for Model Building

Model building methods originated in X-ray crystallography (Cowtan, 2006) and were later adapted for cryo-EM data. Early methods (Emsley & Cowtan, 2004; Cowtan, 2006; Lindert et al., 2009; Baker et al., 2012; Wang et al., 2015; Chen et al., 2016; Terashi & Kihara, 2018; Terwilliger et al., 2018) primarily focus on low-level geometric features, such as density connectivity (Chen et al., 2016; Terashi & Kihara, 2018) and local topology (Al Nasr et al., 2013; Terwilliger et al., 2018). In particular, they typically employ multi-stage pipelines to identify structural representative points (Terashi & Kihara, 2018) derived from low-level geometric features, followed by structural connection or fitting (Terwilliger et al., 2018; Terashi & Kihara, 2018) and iterative refinement (Wang et al., 2015; Terwilliger et al., 2018; Terashi & Kihara, 2018). However, because cryo-EM resolution is often non-uniform, low-level geometric features frequently become unreliable in low-quality regions, resulting in incomplete obtained atomic models and requiring labor-intensive manual correction (Jamali et al., 2024). While our approach leverages large-scale, data-driven learning to extract robust structural patterns that remain physically plausible even in low-quality density regions.

### 2.2. Sequence-based Atomic Model Prediction

On the other hand, sequence-based structure prediction models (Senior et al., 2020; Jumper et al., 2021; Abramson et al., 2024; Wohlwend et al., 2025; Passaro et al., 2025; Meier et al., 2021; Lin et al., 2023; Hayes et al., 2025; Baek et al., 2021; 2024; Ahdritz et al., 2024; Yi et al., 2025; Kagaya et al., 2025), notably AlphaFold (Senior et al., 2020; Jumper et al., 2021; Abramson et al., 2024), have achieved revolutionary breakthroughs by providing atomic-level accuracy in predicting 3D structures directly from sequence information (Kuhlman & Bradley, 2019; Gomes et al., 2022; Genc & McGuffin, 2024; Meng et al., 2025). These methods employ end-to-end deep learning architectures to extract evolutionary information from multiple sequence alignments (MSAs) (Yanofsky et al., 1964; Altschuh et al., 1988; Göbel et al., 1994) and inter-residue spatial (Passaro et al., 2025), enabling high-fidelity structure prediction. However, as these approaches are strictly sequence-driven, they operate without the direct guidance of experimental cryo-EM density maps. This absence of experimental grounding often leads to discrepancies with observed data, particularly when evolutionary information is sparse or structural templates are unavailable (Chowdhury et al., 2022; Pearson, 2013; Perdigão et al., 2015). To overcome these limitations, we present a multimodal framework built upon the open-source Boltz (Wohlwend et al., 2025) architecture, which integrates cryo-EM density maps as an explicit structural modality.

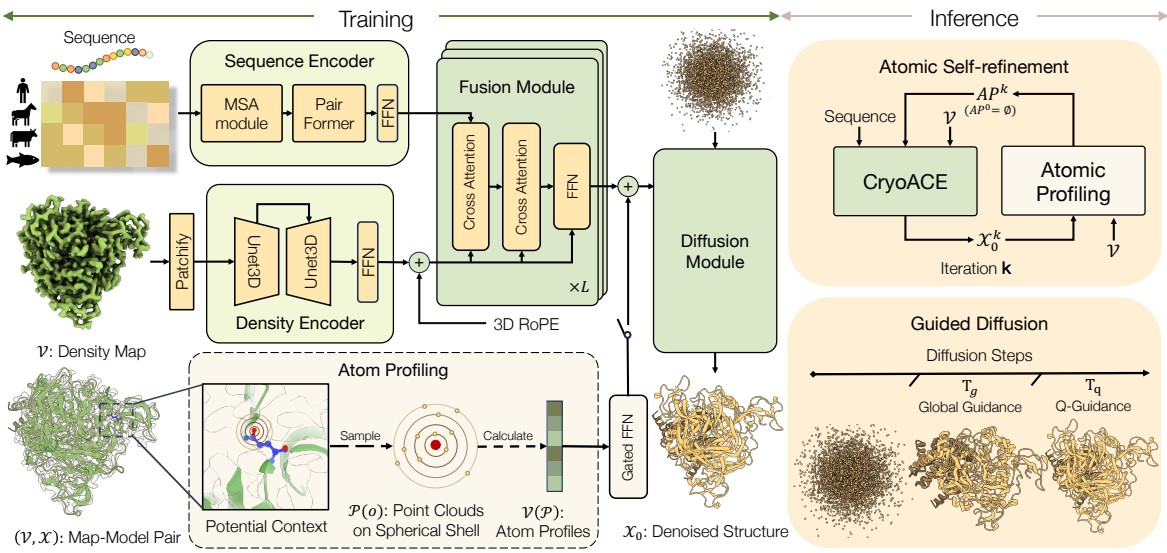

*Figure 1.* **Pipeline of CryoACE.** The pipeline integrates three modalities, including sequences, electron density maps, and atomic profiles, to build atomic structures via a diffusion-based decoder. At the inference stage, we employ an atomic self-refinement strategy. This process is further enhanced by a guided diffusion scheme, utilizing global guidance ($T_g$) and Q-guidance ($T_q$) to improve model performance.

## 2.3. Neural Methods for Automated Model Building

Deep learning approaches (Maddhuri Venkata Subramaniya et al., 2019; Si et al., 2020; Pfab et al., 2021; Zhang et al., 2022; Wang et al., 2023; Terashi et al., 2024; Wang et al., 2024; Terashi et al., 2025) have evolved beyond low-level geometric features, utilizing U-Net architectures (Si et al., 2020; Pfab et al., 2021; Zhang et al., 2022) or hybrid 3D transformer-Hidden Markov Models (Giri & Cheng, 2024). A pivotal shift occurred with EMBuild (He et al., 2022), which introduced sequence information. Building on this, ModelAngelo (Jamali et al., 2024) established a new state of the art by fusing GNNs with ESM sequence embeddings (Lin et al., 2023). Following this trajectory, tools like E3CryoFold (Wang et al., 2025) and CryoAtom (Su et al., 2025) leverage sequence cues from structure prediction networks. Despite these advances, resulting atomic models frequently remain fragmented and lack precision. A parallel emerging trend integrates structure prediction models directly as priors within generative diffusion frameworks. For instance, CryoBoltz (Raghu et al., 2025) adopts a training-free guidance strategy, utilizing the Boltz (Wohlwend et al., 2025) model to steer coordinate generation. While this leverages strong structural priors, the model itself lacks intrinsic recognition of the density modality.

To address these limitations, we propose CryoACE, an atom-centric framework that can accurately and automatically generate atomic models for density maps derived from either homogeneous or heterogeneous cryo-EM datasets.

## 3. Preliminaries

### 3.1. Diffusion-Based Sampling in AlphaFold3

AlphaFold 3 (Abramson et al., 2024) represents a paradigm shift from the structural module of its predecessor by incorporating a generative diffusion model to predict protein coordinates. Formally, let $\mathbf{x}_0 \in \mathbb{R}^{N \times 3}$ denote the ground-truth atomic coordinates of a protein with $N$ atoms. The diffusion process involves a forward stochastic differential equation (SDE) that gradually adds Gaussian noise to the 3D structure over a continuous time duration $t \in [0, 1]$, transforming $\mathbf{x}_0$ into a Gaussian distribution $\mathbf{x}_T \sim \mathcal{N}(\mathbf{0}, \mathbf{I})$. The generative reverse process aims to recover $\mathbf{x}_0$ from $\mathbf{x}_T$ by learning a score function (or a denoiser). The model is conditioned on the Multiple Sequence Alignment (MSA) representations and pair features, denoted as $\mathbf{c}$. The step of denoising process at time $t$ can be parameterized as estimating the noise $\epsilon_\theta(\mathbf{x}_t, t, \mathbf{c})$ or directly predicting the structure $\hat{\mathbf{x}}_0(\mathbf{x}_t, t, \mathbf{c})$. In AlphaFold 3, the training objective focuses on minimizing the difference between the predicted and ground-truth structures, often utilizing a diffusion loss combined with structural validity terms:

$$\mathcal{L}_{\text{diff}} = \mathbb{E}_{t, \mathbf{x}_0, \epsilon} \left[ w(t) \| \hat{\mathbf{x}}_0(\mathbf{x}_t, t, \mathbf{c}) - \mathbf{x}_0 \|^2 \right] \quad (1)$$

where $w(t)$ is a weighting function. This diffusion-based approach allows the model to sample diverse structural conformations and model distribution uncertainty effectively.

### 3.2. Forward Model of Cryo-EM Map Simulation

The generation of a cryo-EM density map from atomic coordinates can be formulated as a forward physical model. An electron density map $V : \mathbb{R}^3 \to \mathbb{R}$ represents the elec-

trostatic potential distribution of the molecule.

Given a set of atomic coordinates $\mathbf{X} = \{\mathbf{r}_1, \ldots, \mathbf{r}_N\}$ and their corresponding atomic numbers (or types) $\{Z_1, \ldots, Z_N\}$, the ideal continuous electron density $\rho(\mathbf{r})$ is commonly approximated as a superposition of Gaussian functions centered at each atom. The density at a spatial location $\mathbf{r}$ is defined as:

$$\rho(\mathbf{r}) = \sum_{i=1}^{N} A_i \exp\left(-\frac{\|\mathbf{r} - \mathbf{r}_i\|^2}{2\sigma_i^2}\right) \quad (2)$$

where $A_i$ is the amplitude related to the atomic mass and scattering factor, and $\sigma_i$ controls the width of the Gaussian, which is determined by the resolution of the map and the B-factor (thermal displacement parameter) of the atom.

In practice, this continuous function is discretized onto a 3D voxel grid $V \in \mathbb{R}^{D \times H \times W}$. For a voxel $\mathbf{v}_k$, the intensity is given by the integration or sampling of $\rho(\mathbf{r})$ within that voxel. This process allows us to simulate a theoretical density map $\hat{V}$ from any predicted structure $\hat{\mathbf{x}}$, enabling direct comparison with experimental cryo-EM volumes.

## 4. Methods

As illustrated in Figure 1, our atom-centric multi-modal framework, CryoACE, utilizes a cross-attention fusion module to aggregate features from separated sequence and density embeddings, added by local atom profiling biases to guide coordinate synthesis (Section 4.1). The model is trained on map-model pairs to learn effective denoising trajectories (Section 4.2), while inference progressively reconstructs the structure using an atomic-recycling self-refinement strategy and Guided Diffusion (Section 4.3).

### 4.1. Model Architecture

**Sequence encoding.** For a complex comprising $M$ chains with a total of $N$ residues, we first construct the corresponding multiple sequence alignments (MSAs) (Yanofsky et al., 1964). These evolutionary data are processed through a Boltz-1-style encoder (Wohlwend et al., 2025), consisting of a specialized MSA module and a Pairformer block. Specifically, the encoder transforms the raw MSAs into high-dimensional sequence tokens $\mathbf{f}_s \in \mathbb{R}^{N \times D}$:

$$\mathbf{f}_s = \text{PairFormer}(\text{MSAmodule}(\mathcal{S})) \quad (3)$$

where $\mathcal{S}$ is the input sequence and MSA pairs, provides a robust semantic backbone that guides the spatial positioning of amino acid residues within the cryo-EM density map.

**Density encoding.** The density encoder extracts spatial features from the volumetric density map $\mathcal{V} \in \mathbb{R}^{l \times l \times l}$. First, $\mathcal{V}$ is uniformly partitioned into $K$ non-overlapping 3D patches $\{\mathcal{V}^i\}_{i=1}^{N}$. These patches are individually mapped to latent embeddings via a 3D ResUNet (Çiçek et al., 2016) and

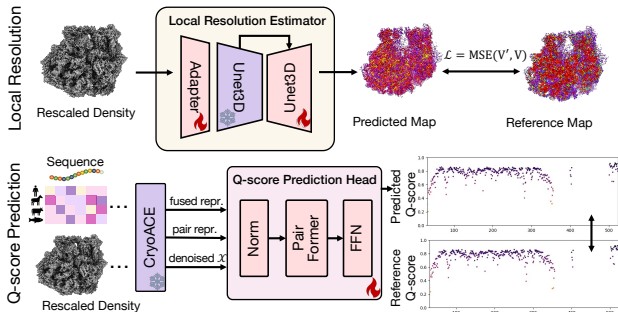

*Figure 2.* **Light-weight downstream heads**. The architecture consists of two specialized modules: a local resolution estimator (top) that uses a Unet3D to predict density maps from rescaled inputs, and a Q-score prediction head (bottom) that uses embeddings from a frozen CryoACE backbone to estimate residue-level quality.

concatenated to form a patch token sequence $\mathbf{f}_d \in \mathbb{R}^{N \times D}$:

$$\mathbf{f}_d = \text{Concat}\left(\text{UNet3D}(\mathcal{V}^1), \ldots, \text{UNet3D}(\mathcal{V}^K)\right) \quad (4)$$

Before these features are passed into the fusion module, we apply 3D Rotary Positional Embeddings (3D RoPE) (Ma et al., 2025) to the token sequence $\mathbf{f}_d$ based on the 3-dimensional spatial index of each patch.

**Atom profiling.** To bridge the gap between global volumetric data and discrete atomic coordinates, we introduce an atomic profiling module that extracts localized density descriptors. For each atom $i$ at position $\mathbf{x}_i$, the module samples the local density environment from the volumetric map $\mathcal{V}$ via trilinear interpolation:

$$\mathbf{p}_i = \phi\left(\text{interp}(\mathcal{V}, \mathbf{x}_i)\right) \quad (5)$$

where $\text{interp}(\cdot)$ queries the density value at the continuous coordinate $\mathbf{x}_i$, and $\phi$ is a learnable projection that transforms these local intensities into a high-dimensional descriptor. The physical intuition is straightforward: in cryo-EM, the Coulomb potential peaks exactly at atomic positions, so sampling the volumetric map at predicted atomic coordinates yields the most informative local signal. This is also what enables effective self-refinement, as improved predicted coordinates across iterations produce more accurate density readouts, feeding better features into the next refinement cycle.

**Fusion module.** We adopt a transformer-based module that integrates structural features from the density encoder into the sequence-based backbone. We employ $L$ layers of cross-attention block, where sequence tokens act as queries $(Q)$ to retrieve relevant spatial features from the density tokens $(K, V)$:

$$\mathbf{f}_{agg} = \text{CrossAttention}(\mathbf{f}_s, \mathbf{f}_d, \mathbf{f}_d) \quad (6)$$

By querying density tokens with sequence embeddings, the module imposes evolutionary constraints onto the volumetric data, ensuring structural consistency even in low resolution regions.

**Diffusion Module.** The diffusion module acts as the structural decoder, generating final atomic coordinates $\mathbf{x} \in \mathbb{R}^{3N}$ through a conditional reverse diffusion process. Given the fused features $\mathbf{f}_{fused}$, the model learns to map a noise-perturbed configuration $\mathbf{x}_t$ back to the clean structural manifold at each timestep $t$:

$$\hat{\mathbf{x}}_0 = \text{Atom-Transformer}(\mathbf{x}_t, t, \mathbf{f}_{fused}) \qquad (7)$$

We implement the denoiser using an atom-transformer-based architecture (Abramson et al., 2024), which employs self-attention to capture multi-scale spatial dependencies and inter-residue interactions.

**Light-weight downstream heads.** As shown in Fig. 2, to extend CryoACE capabilities, we introduce two auxiliary modules. The local resolution estimator captures map resolvability by processing rescaled densities through an adapter and 3D U-Net trained with MSE loss. Parallelly, the Q-score prediction head estimates residue-level confidence by leveraging embeddings from the frozen CryoACE backbone. These features are regressed into atomic Q-scores passing through prediction heads based on Boltz-1 confidence module, enabling efficient quality assessment without heavy computation. Refer to Appendix B for more details.

### 4.2. Training

**Conditional Dropout.** To synchronize the multi-modal integration and handle the absence of atomic profiles during the initial stage, we employ a conditional dropout strategy. Specifically, during training, the atomic profile $\mathbf{f}_{ap}$ is randomly replaced by a null vector $\emptyset$ with a probability $p$. This approach forces the model to learn meaningful representations from sequences and density maps alone, while effectively utilizing profile information when available.

**Training objective.** The overall training objective of CryoACE is to minimize a multi-task loss function that ensures both structural plausibility and density-map fidelity. The primary component is the **diffusion loss**, which supervises the denoising of atomic coordinates $\mathbf{x}$:

$$\mathcal{L}_{diff} = \mathbb{E}_{\mathbf{x}_0, \epsilon, t} \left[ \|\hat{\mathbf{x}}_0(\mathbf{x}_t, t, \mathbf{f}_{fused}) - \mathbf{x}_0\|_2^2 \right] \qquad (8)$$

where $\mathbf{x}_0$ represents the ground-truth atomic positions. To further enforce the consistency between the generated model and the cryo-EM density, we incorporate a map-fitting loss $\mathcal{L}_{fit}$. This term is defined as the mean squared error between the simulated density of the predicted atoms and the experimental density map $\mathcal{V}$:

$$\mathcal{L}_{fit} = \|\mathcal{V}_{\text{sim}}(\hat{\mathbf{x}}_0) - \mathcal{V}\|_2^2 \qquad (9)$$

where $\mathcal{V}_{sim}(\hat{\mathbf{x}}_0)$ is the density map simulated from the denoised atomic coordinates $\hat{\mathbf{x}}_0$, and $N$ is the total number of voxels. The total training objective is formulated as:

$$\mathcal{L}_{total} = \lambda_1 \mathcal{L}_{diff} + \lambda_2 \mathcal{L}_{fit} + \lambda_3 \mathcal{L}_{aux} \qquad (10)$$

where $\mathcal{L}_{aux}$ integrates auxiliary structural priors from Boltz-1, such as FAPE (Frame Aligned Point Error) and stereochemical constraints (bond lengths and angles). This joint optimization allows CryoACE to capture fine-grained atomic details while maintaining strict alignment with the experimental density-map constraints.

### 4.3. Inference

During inference, CryoACE generates the atomic model through a multi-stage iterative process that combines generative sampling with structural refinement. Unlike training, where ground-truth profiles are available, the inference phase starts from a sequence-and-density-only state and progressively incorporates structural feedback to achieve high-fidelity reconstruction.

**Atomic self-refinement.** To bridge the modality gap at test time, we introduce an atomic self-refinement strategy. We initialize the atomic profiles $AP^0$ as an empty set, allowing the model to produce an initial coarse structure $\mathbf{x}^0$ based on sequence and density features. In each subsequent iteration $k$, the Atomic Profiling module takes previously generated structure $\mathbf{x}^{k-1}$ as input to compute updated profiles $AP^k$ to generate the refined $\mathbf{f}_{ap}^k$ that is added into the fused feature as control signal. This feedback loop enables the model to refine local atomic environments dynamically, effectively correcting misplacements and completing missing residues.

**Training-free guidance.** To achieve accurate automated modeling, we introduce a multi-guidance framework that drives the diffusion process across different map resolutions. Following cryoboltz (Raghu et al., 2025), in the initial stages of denoising, we employ global guidance by transforming the cryo-EM density map into a compact 3D point cloud via weighted k-means, we compute the Sinkhorn divergence to efficiently align the model's global topology with the target density map. As the denoising transitions into its terminal phases, the focus shifts from global morphology to local refinement via Q-score guidance.

$$Q_i = \text{Corr}\left( \text{interp}(\mathcal{V}, \mathbf{x}_i + \delta), \exp\left(-\frac{|\delta|^2}{2\sigma^2}\right) \right) \qquad (11)$$

This coarse-to-fine strategy ensures that the generated structures maintain both correct global assembly and high-fidelity atomic details.

We note that CryoACE is intentionally modular: components such as self-refinement, atom profiling, and the auxiliary guidance heads can be selectively disabled to prioritize

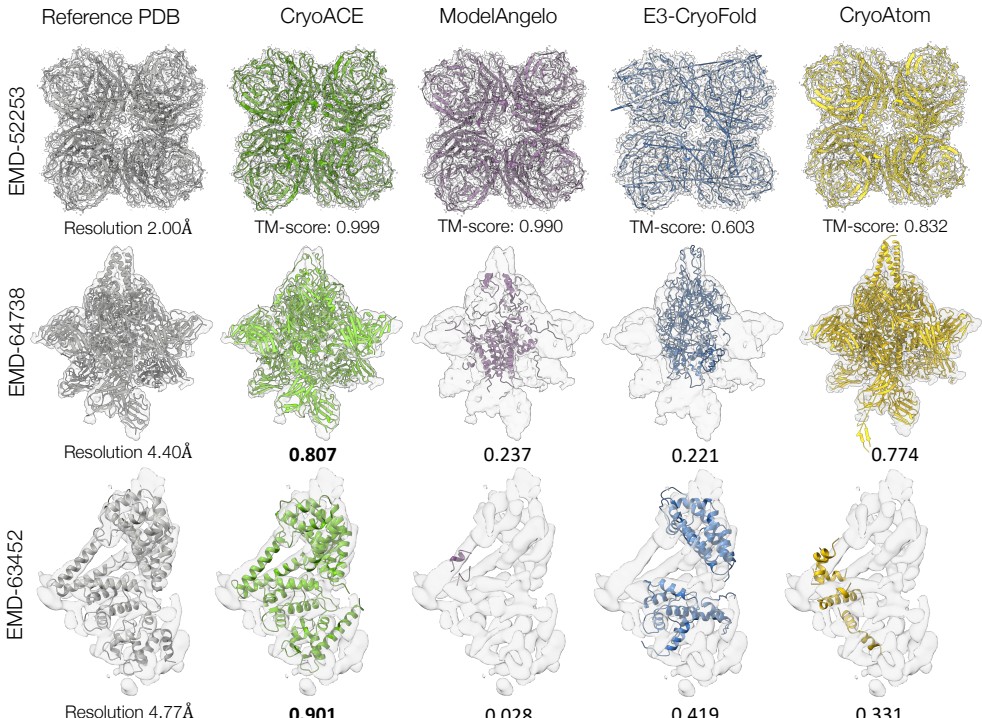

*Figure 3.* **Qualitative comparison of automated model building from homogeneous density maps.** We demonstrate three examples with resolution values decreasing from top to bottom. While CryoACE and ModelAngelo achieve near-perfect TM-scores on the 2.00 Å map, E3-CryoFold suffers from backbone breaks. Crucially, in the lower resolution cases, only CryoACE generates complete, high TM-score structures. This highlights the robustness of CryoACE, whereas other methods (particularly those sensitive to the 4 Å threshold) fail to produce valid models.

throughput when structural features are unambiguous (e.g., high-resolution maps), while the full pipeline provides the largest accuracy gains in challenging low-resolution regimes (3.5–4.0 Å).

## 5. Experiments

**Implementation details.** CryoACE was trained for 500 epochs on 32 NVIDIA H20 GPUs using the AdamW optimizer (learning rate $1 \times 10^{-4}$, cosine annealing decay). The conditional dropout for atomic profiles was set to $0.2$, ensuring the model maintains high performance even when local density cues are noisy. We employed a two-stage strategy: the Boltz-1 initialized sequence and structure modules were frozen for the first 100 epochs to allow the 3D ResUnet and the fusion module to adapt to the density map modality, followed by end-to-end fine-tuning.

**Training dataset.** A large-scale, high-quality, and accurately aligned training dataset is critical for effective cryo-EM multi-modal modeling. However, raw entries from public repositories such as electron microscopy data bank (EMDB) (Lawson et al., 2016), and protein data bank (PDB) (Burley et al., 2021) frequently suffer from severe spatial misalignments and structural incompleteness. To ad-

dress this, we curated a large-scale yet high-quality dataset of 10,915 curated density-structure-sequence triplets, distilled from an initial pool of 13,775 candidates (released before 2025, $< 4$ Å). Our novel curation pipeline ensures data integrity through dual-stage alignment and strict quality-based filtering based on Q-scores and model completeness. Additionally, we computed local resolutions to train our auxiliary head for local resolution estimation. To ensure fair comparison, we strictly excluded the Cryo2StructData (Giri et al., 2024) test set. Details in Appendix A.2.

**Ablation study.** We perform a comprehensive component-wise ablation on the homogeneous benchmark. Every component contributes positive and consistent gains, confirming the non-redundancy of our architecture. The two largest contributors are the learned density encoder (+4.8% BB Acc.) and atomic self-refinement (+3.6%), validating density conditioning during training and iterative refinement at inference as the central design pillars; atom profiling (+1.8%), auxiliary heads (+1.1%), and global guidance (+0.7%) each provide incremental but consistent gains.

Detailed analyses and additional dataset/inference-strategy ablations are provided in Appendix D.

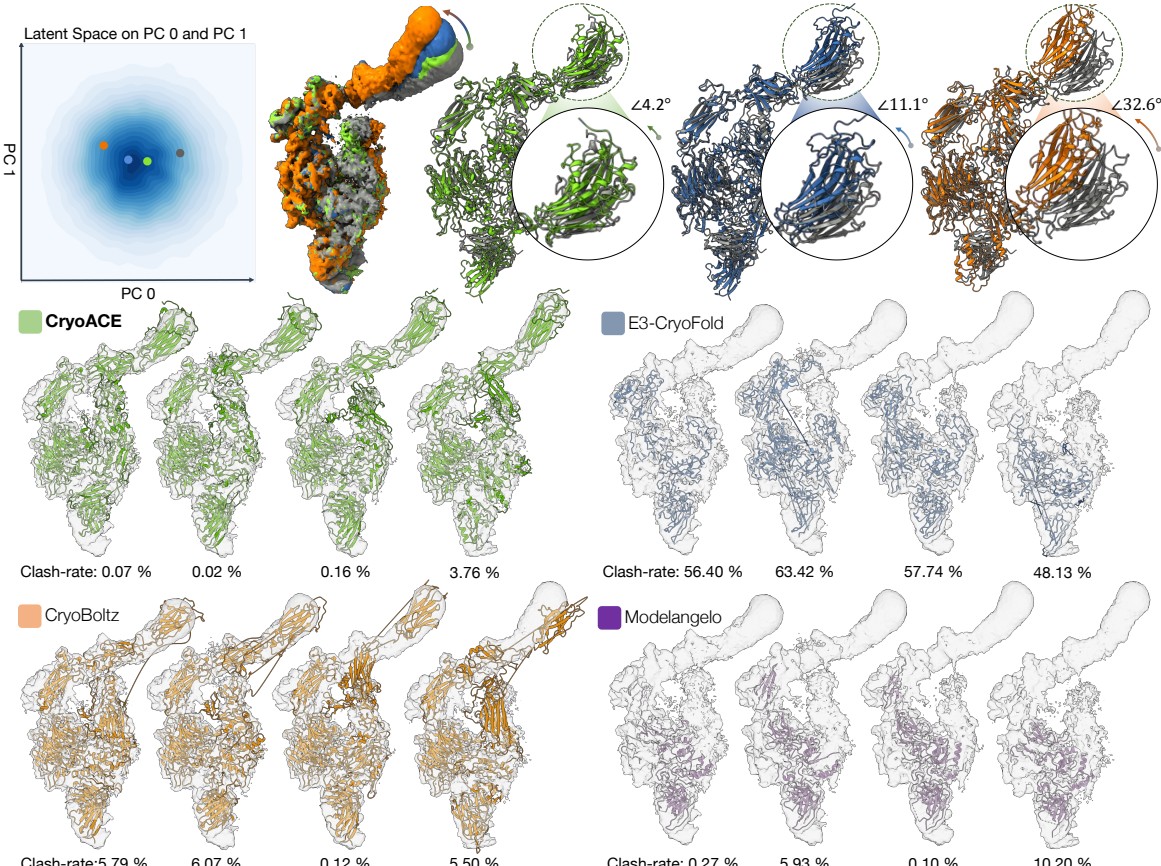

*Figure 4.* **Comparison of Heterogeneous Model Building Results on the $\alpha V\beta 8$ Integrin.** This target exhibits significant flexibility in the Fab arm. Existing methods like E3-CryoFold and ModelAngelo fail to model this dynamic region, while CryoBoltz attempts a full reconstruction but suffers from backbone breaks and visual misalignment. In contrast, CryoACE generates a complete, continuous structure that accurately fits the density map, even for the highly dynamic Fab arm. The **top-left panel** illustrates the reconstructed conformational latent space, where four marked points correspond to the sampled conformations shown.

*Table 1.* **Quantitative comparison for Homogeneous Model Building.**

| METHODS | COMPLETENESS | BB ACCURACY | AA ACCURACY | BB RMSD (Å) | AA RMSD (Å) | Q-SCORE |
|---|---|---|---|---|---|---|
| E3-CRYOFOLD | 78.21% | 86.5% | - | 3.4 | - | 0.433 |
| CRYOATOM | 91.54% | 94.6% | 82.1% | 2.3 | 3.5 | 0.521 |
| MODELANGELO | 89.17% | 93.7% | 84.3% | 2.7 | 3.3 | 0.513 |
| OURS | **100%** | **98.3%** | **87.5%** | **2.1** | **3.0** | **0.524** |

*Table 2.* **Quantitative Comparison on Experimental Heterogeneous Datasets.**

| EMPIAR-ID | METHODS | CLASH RATE | MOLPROBITY | COMPLETENESS | WEIGHTED CC | PSC |
|---|---|---|---|---|---|---|
| 10345 | E3-CRYOFOLD | 39.443% | 2.830 | 58.3% | 0.316 | 0.362 ± 0.088 |
| | MODELANGELO | 2.359% | 1.997 | 28.8% | 0.217 | 0.681 ± 0.061 |
| | CRYOBOLTZ | 2.363% | 1.365 | **100%** | 0.641 | 0.520 ± 0.088 |
| | CRYOACE | **0.446%** | **1.198** | **100%** | **0.642** | 0.526 ± 0.051 |
| 10516 | E3-CRYOFOLD | 25.206% | 2.760 | 73.1% | 0.382 | 0.377 ± 0.032 |
| | MODELANGELO | 1.487% | 2.173 | 57.4% | 0.334 | 0.834 ± 0.134 |
| | CRYOBOLTZ | 6.475% | **1.403** | **100%** | 0.636 | 0.904 ± 0.032 |
| | CRYOACE | **1.102%** | 1.480 | **100%** | **0.642** | 0.909 ± 0.021 |

## 5.1. Homogeneous Model Building

We evaluate CryoACE on the task of automated atomic structure modeling from static homogeneous cryo-EM density maps. The objective is to derive precise atomic coordinates directly from volumetric inputs.

**Baselines.** We benchmark CryoACE against three state-of-the-art neural approaches including **ModelAngelo** (Jamali et al., 2024), **CryoAtom** (Su et al., 2025), and **E3-CryoFold** (Wang et al., 2025), as well as two sequence-only foundation-model baselines, **Boltz-1** (Wohlwend et al., 2025) and **CryoBoltz** (Raghu et al., 2025), to disentangle the contributions of inference-time density guidance versus learned density conditioning. We evaluated them using their official implementations with default configurations. All experiments are conducted on the public **Cryo2StructData** test set, which was strictly excluded from our training dataset to ensure zero data leakage.

**Metrics.** We evaluate model quality using a comprehensive set of metrics covering three key dimensions: geometric accuracy, structural completeness, and density fidelity. Specifically, we report both root mean square deviation (**RMSD**) and **Accuracy** (percentage of atoms within $3.0\,\text{Å}$) at coarse-grained backbone (**BB**) and fine-grained all-atom (**AA**) levels to quantify global spatial deviation from the deposited reference structure. **Completeness** is measured as the ratio of built residues to the ground truth sequence. Furthermore, we utilize the **Q-score** to evaluate the local correlation between atomic coordinates and the input density map, ensuring that the predicted structures maintain high consistency with the experimental evidence.

**Results.** Figure 3 demonstrates the model building fidelity on three representative test cases ranging from $2.00\,\text{Å}$ to $4.77\,\text{Å}$ (released after Jan 1, 2025; unseen during training for all method). In the high-resolution setting: the **Influenza Neuraminidase** complex (EMD-52253, $2.00\,\text{Å}$) (Moran et al., 2025), CryoACE achieves near-optimal alignment with the ground truth (TM-score 0.999), effectively resolving fine-grained atomic details. While ModelAngelo remains competitive in this ideal case, E3-CryoFold and CryoAtom exhibit observable structural artifacts.

Crucially, in challenging low-resolution conditions ($>$ $4.0\,\text{Å}$) like the **RSV pre-F trimer** (EMD-64738) (Zhai H, 2025) and the large cytosolic **CatSpermasome subcomplex** (EMD-63452) (Zhao et al., 2025), baselines struggle with noisy density maps, yielding fragmented structures with TM-scores $< 0.45$. In contrast, CryoACE maintains superior topological fidelity. This resilience is driven by our atom profiling mechanism to efficiently capture fine-grained local geometric cues to guide the diffusion process.

As shown in Table 1, we also conduct a comprehensive quantitative evaluation on the Cryo2StructData test set.

CryoACE demonstrates superior geometric fidelity with the highest accuracy and minimized RMSD levels in both backbone and all-atom levels. Furthermore, our method achieves the best Q-score, implying that the generated models maintain good agreement with the experimental density maps. Finally, attributed to our atomic-centric design, CryoACE guarantees full sequence coverage, achieving nearly 100% structural completeness. This stands in sharp contrast to baseline approaches, which all suffer from varying degrees of partial modeling and fragmentation. Note that we do not report the all-atom accuracy and RMSD metric for E3-CryoFold as it cannot generate meaningful results that can pass its default postprocessing procedure.

## 5.2. Heterogeneous Model Building

We extend our evaluation to the task of modeling dynamic structures from two ensembles of reconstructed density maps. The objective is to derive distinct atomic coordinates for each conformational state. Unlike the homogeneous setting, ground truth PDB structures are unavailable, requiring density-based validation.

**Experimental Datasets.** To evaluate our model's capability on challenging real-world data exhibiting significant heterogeneity, we selected two representative datasets: the **asymmetric $\alpha V \beta 8$ Integrin** (EMPIAR-10345) (Campbell et al., 2020) and the SARS-CoV-2 Spike protein (EMPIAR-10516) (Melero et al., 2020). For the Integrin complex, we utilized particle stacks provided by CryoDRGN (Zhong et al., 2021). For the Spike protein, we employed the standard CryoSPARC pipeline to extract a high-quality particle stack from raw movies hosted on EMPIAR (Iudin et al., 2016). Subsequently, we applied the default CryoDRGN framework to both targets to model their continuous conformational latent spaces, enabling the reconstruction of diverse heterogeneous density maps at full resolution. Finally, to curate a representative evaluation set, we performed Gaussian Mixture Model (GMM) clustering within the latent space to generate 10 distinct density maps for each target. Note that we manually removed the density signals of a Fab fragment from the Integrin maps, as its sequence is unavailable. This preprocessing prevents baseline methods from being penalized for failing to reconstruct unsequenced regions.

**Metrics.** To evaluate the heterogeneous models, we assess four key dimensions: physical plausibility, structure completeness, map-model fitness, and conformational consistency. Physical plausibility is measured by the **Clash rate** (atomic overlaps) and the **MolProbity score** (Chen et al., 2010), which integrates clashscore (serious clash), rotamer and Ramachandran validations. Completeness is defined as the ratio of predicted residues to the ground truth sequence length. For map-model fitness, we report a **Weighted Cross-Correlation (WCC)**, which weights the standard correla-

tion by a volumetric occupancy factor. Finally, we evaluate the generated ensemble via the **Pairwise Structural Consistency (PSC)**, the average TM-score between adjacent frames. We target a PSC $> 0.5$ with low variance to ensure structural stability, while the upper bound is protein-specific. Protein with more rigid domains are expected to exhibit higher PSC upper bounds. Details of the definition for metrics are in Appendix C

**Results.** As shown in Table 2, our method performs the best among all models. We achieve significantly better physical quality than E3-CryoFold and ModelAngelo. We also outperform CryoBoltz, thanks to our map-structure co-training and effective coarse-to-fine sampling strategy. Unlike E3-CryoFold and ModelAngelo, which often fail in low-resolution regions, our approach ensures full-sequence reconstruction with higher completeness and much better wCC scores. Notably, ModelAngelo's high PSC score on EMPIAR-10345 is misleading because it only models the 30% rigid core. In contrast, our method maintains a stable TM-score $> 0.5$ across the entire ensemble, demonstrating reliable and consistent performance throughout the reconstruction process. The improvement over CryoBoltz on low-SNR cases (e.g., RSV-preF, ARMH2) stems from a fundamentally different design philosophy: CryoBoltz derives gradients from the raw density map at inference, which is reliable at high resolution but becomes weak and noisy in low-SNR regimes, whereas CryoACE trains its density encoder on maps spanning a wide resolution range (2–4 Å), so the model learns to extract robust structural features from noisy observations during training rather than relying on test-time gradient signals. We note that publicly available benchmarks for heterogeneous atomic model building remain scarce, and our evaluation here therefore covers the largest curated set we are aware of rather than an exhaustive coverage of all possible scenarios.

## 6. Discussion

**Limitations.** Despite promising results, CryoACE does have several limitations. First, our training dataset currently lacks nucleic acid complexes. Consequently, DNA/RNA prediction accuracy relies on pre-trained Boltz priors rather than learned representations from our pipeline. Second, CryoACE incurs significant GPU memory costs when processing long sequences, restricting the inference of large macromolecular assemblies on standard devices. Finally, although our method improves robustness on lower-resolution maps, performance degrades at resolutions coarser than 8Å. Enhancing accuracy in these challenging cases remains a key direction for future research.

Quantitatively, on the low-resolution CYP102A1 open state (6.5 Å), CryoACE achieves a C$\alpha$ RMSD of 2.80 Å, substantially worse than the typical sub-Å accuracy at high resolution. Non-protein components remain out of scope:

on the Ro60/La/pre-5S rRNA complex (PDB 9NFA, 2.7 Å), CryoACE attains a TM-score of 0.92 on protein chains but exceeds 4 Å RMSD on the RNA component. GPU memory also becomes a bottleneck for sequences longer than ∼2000 residues on a single A800 GPU.

**Conclusion.** In this work, we have presented CryoACE, an atom-centric generative framework designed to construct physically plausible atomic graphs directly from both homogeneous and heterogeneous cryo-EM density maps. By seamlessly integrating an efficient coarse-to-fine density sampling strategy with a novel training-free energy guidance mechanism, our model effectively bridges the gap between raw volumetric observations and strict biochemical constraints. Extensive evaluations demonstrate that CryoACE establishes new state-of-the-art performance on standard benchmarks while successfully resolving complex dynamic ensembles in challenging real-world heterogeneous datasets. We believe that CryoACE will serve as a useful tool for high-throughput structural biology.

## Acknowledgement

This work was supported in part by National Key RD Program of China 2025YFA1309603, the National Natural Science Foundation of China under Grant W2431046, Central Guided Local Science and Technology Foundation of China YDZX20253100001001, and by MoE Key Lab of Intelligent Perception and Human-Machine Collaboration (ShanghaiTech University), the Shanghai Frontiers Science Center of Human-centered Artificial Intelligence. The experiments of this work were supported by the SIST computing Platform and HPC, ShanghaiTech University.

## Impact Statement

This paper presents CryoACE, a method aimed at automating atomic structure modeling from cryo-EM data. The primary societal goal of this work is to accelerate structural biology research, which plays a critical role in understanding disease mechanisms and facilitating structure-based drug discovery. By resolving complex heterogeneous conformations, our tool has the potential to lower the barrier for analyzing difficult macromolecular assemblies.

However, we acknowledge potential risks associated with generative models in scientific contexts. Specifically, there is a risk of "hallucinating" plausible but incorrect atomic structures, which could mislead downstream biological analysis or drug design efforts if not rigorously validated. We encourage users to treat the model's outputs as hypotheses that require experimental verification. We do not foresee significant negative societal consequences or immediate dual-use concerns beyond the general necessity for responsible deployment of AI tools in biomedicine.

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

# A. Dataset

## A.1. Imperfections of public datasets

Although the EMDB and PDB repositories provide vast amounts of raw data, directly utilizing them for multi-modal learning is non-trivial. This is because raw entries frequently exhibit four major issues: spatial misalignments between atomic coordinates and density maps, mismatches between full-length sequences and partially modeled structures, incomplete atomic models (e.g., viral assemblies), and low volumetric occupancy where the target structure is surrounded by excessive background noise.

While recent initiatives like Cryo2StructData (Giri et al., 2024) have attempted to standardize this data, they remain constrained by significant limitations. First, Cryo2StructData is temporally outdated, with a data cutoff in March 2023, effectively omitting a wave of recent high-resolution structures. Second, and more critically, the dataset construction was limited to automated preprocessing without rigorous data cleaning. A detailed inspection reveals that a non-negligible portion of the data is still compromised by severe map-model misalignments and corrupted volume files, the latter may be caused by the non-robust preprocessing pipeline. Furthermore, many entries contain unmodeled density regions that lack corresponding atomic coordinates. Training on such noisy data introduces erroneous supervision signals, which can predispose generative models to hallucinations and convergence instability.

## A.2. Details of data curation

To address these challenges, we curated a refined dataset of 10,915 high-quality triplets, distilled from an initial pool of 13,775 candidates (released prior to January 1, 2025, with reported resolutions better than $4\,\text{Å}$ via a novel curation workflow. The workflow ensures data integrity through a two-stage process. First, we performed two alignments. Structure-Map Alignment aligned synthetic maps generated from CIF coordinates to experimental maps, resulting in significant improvements of Q-score in some misalignment entries. In parallel, Sequence-Structure Alignment was enforced via Clustal Omega; residues present in the sequence but unmodeled in the structure were masked with gap tokens.

Second, we implemented a multi-stage quality filtering pipeline. Initially, we discarded entries with alignment Q-scores below 0.05. Subsequently, to filter out triplets with significant unmodeled region, we introduced the structure modeled rate (**SMR**) as follows:

$$SMR = \frac{V_{sim}(\tau = 0.01)}{V_{exp}(\tau = \tau_{rec})} \tag{12}$$

where $V_{sim}$ is the number of voxels in the simulated density map simulated from the reported resolution, and $V_{exp}$ is the voxel count of the experimental ground truth map. Here, $\tau$ represents the threshold applied to each map, with $\tau_{rec}$ being the author-recommended value. Entries with an **SMR** below 1.5 were excluded to ensure multi-data alignment.

## A.3. Local Resolution Dataset

We also collected raw half-map pairs from EMDB to calculate local resolution using CryoSPARC (Punjani et al., 2017) after full-map and half-map alignment. For entries lacking deposited half-maps in the EMDB, we employed ResMap (Kucukelbir et al., 2014) to estimate local resolution directly from the full-maps.

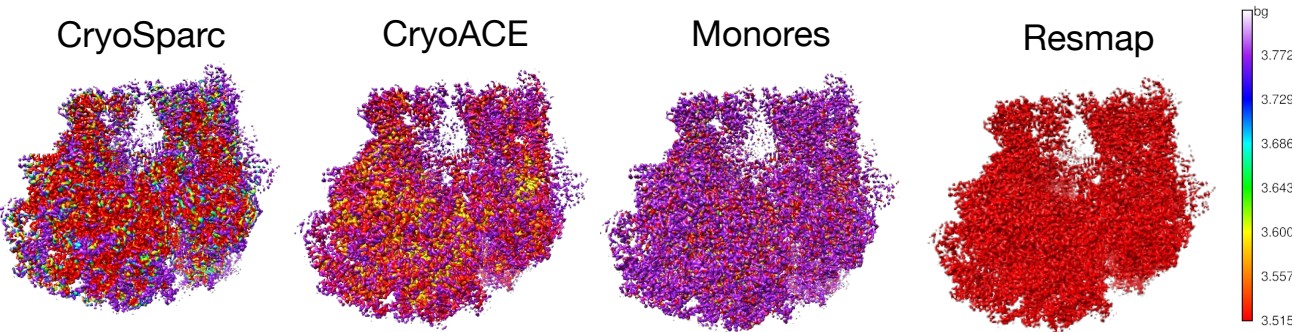

*Figure 5.* **An example of results of local resolution**. Local resolution maps are calculated via CryoRes and visualized through Chimera on case EMD-3492.

# B. Local Resolution Estimation

Standard local resolution is typically derived from the Fourier Shell Correlation (FSC) computed between two independent half-maps, quantifying the resolvability of density features across the 3D volume. In our evaluation, we utilize the local resolution maps generated by CryoSPARC, based on this standard half-map protocol, as the ground truth. We benchmarked our approach against existing methods, specifically ResMap and MonoRes. The results demonstrate that our estimated resolution maps exhibit a significantly higher correlation with the CryoSPARC ground truth than the baselines.

# C. Explanation for metrics

### C.1. Weighted Cross-Correlation (WCC)

We assess model-density agreement using Weighted Cross-Correlation (WCC). This metric is defined as the product of the standard cross-correlation coefficient (CC) and a volume coverage factor. The CC measures the local correlation between the experimental map and the map simulated from the predicted model (computed via ChimeraX). Meanwhile, the coverage factor quantifies the global occupancy, calculated as the ratio of simulated voxels above a minimal density threshold to the experimental voxels above a specific contour level. This ensures the model maximizes correlation while sufficiently filling the target volume.

### C.2. Pairwise Structural Consistency (PSC)

We employ the Pairwise Structural Consistency (PSC) metric to evaluate the temporal coherence and structural stability of the generated conformational ensembles. Defined as the average Template Modeling score (TM-score) between consecutively adjacent frames, PSC assesses how well the topological fold is preserved throughout the dynamic trajectory. We posit that a PSC consistently exceeding 0.5 is the minimal requirement for physical plausibility, ensuring that the global structure does not disintegrate or hallucinate between steps. Additionally, a low variance in these scores is desirable to verify the smoothness of transitions. Crucially, the optimal PSC upper bound is intrinsic to the protein's nature rather than a fixed universal standard. While rigid complexes are expected to exhibit high consistency scores, proteins with significant flexibility naturally yield lower PSC values. Thus, a valid heterogeneous model must balance topological stability with the necessary geometric deviations required to capture true conformational dynamics.

# D. Ablation studies

*Table 3.* Ablation study on the impact of dataset effectiveness and inference strategies.

| METHODS | BB ACCURACY | AA ACCURACY | BB RMSD (Å) | AA RMSD (Å) | Q-SCORE |
|---|---|---|---|---|---|
| C2S DATA | 94.2% | 82.7% | 2.4 | 3.4 | 0.503 |
| C2S DATA + S.R. | 97.8% | 85.2% | 2.2 | 3.2 | 0.521 |
| ACE DATA + S.R. | 98.1% | 87.3% | 2.1 | 3.2 | 0.522 |
| ACE DATA + S.R. + Q-GUIDANCE | **98.3%** | **87.5%** | **2.1** | **3.0** | **0.524** |

**Design choices.** To validate the effectiveness of our proposed inference-time strategies, we conducted an ablation study focusing on the Self-Refinement (S.R.) module and Q-guidance mechanism. As shown in Table 3, the integration of S.R. yielded a substantial performance improvement, increasing backbone accuracy from 94.2% to 97.8% and reducing RMSD significantly. Additionally, introducing Q-guidance served as a final polishing phase, offering marginal gains in precision. These results demonstrate that S.R. progressively enhances structural fidelity through iterative refinement, while Q-guidance further optimizes accuracy by aligning with the density map during the final sampling step.

**Customized dataset.** To validate the effectiveness of our customized dataset, we conducted an ablation study comparing the proposed CryoACEdata (**ACE DATA**) against the Cryo2Structdata (**C2S DATA**). As shown in the second and third line of Table 3, the model trained on ACE DATA consistently outperformed the baseline across all evaluation metrics. We attribute these comprehensive improvements to the expanded scale of CryoACE data and the implementation of our rigorous quality-based filtering pipeline, which collectively mitigate noise and facilitate more robust feature learning.

# E. Additional results

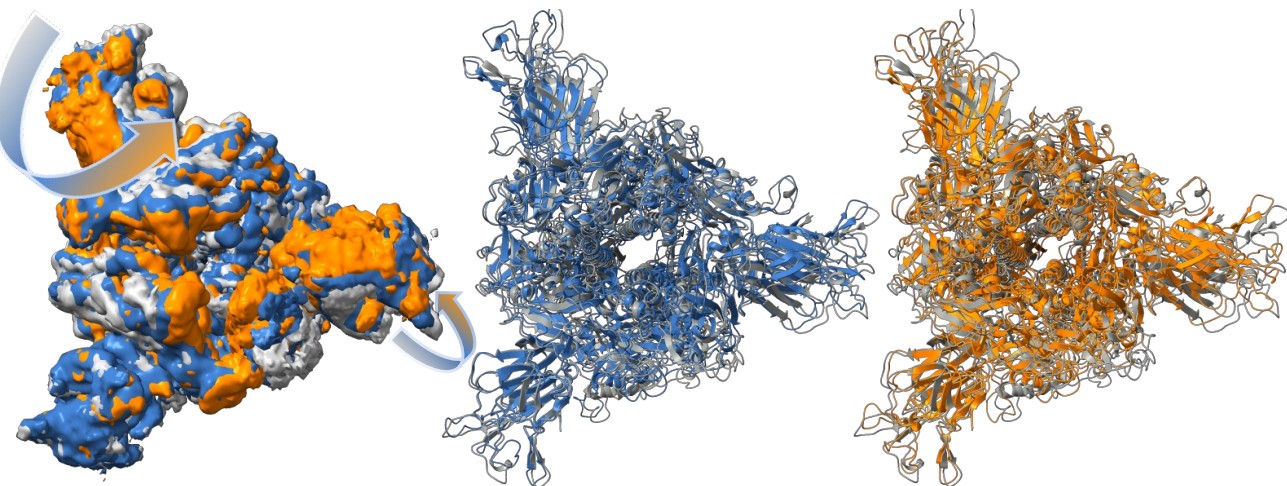

*Figure 6.* **CryoACE's Heterogeneous Model Building Results on SARS-CoV-2 spike in prefusion state (EMPIAR-10516)**

# F. Implementation Details

**Inference.** We perform inference by solving the Probability Flow ODE derived from the linear interpolation $x_t = (1-t)x_0 + t\epsilon$ with $\epsilon \sim \mathcal{N}(0, I)$. Specifically, we employ a first-order Euler solver with 200 steps on a linear schedule $t \in [0, 1]$ to integrate the velocity field from $t=1$ to $t=0$. Each iteration of the self-refinement loop runs a full 200-step diffusion sampling process, and we use $k=3$ such iterations at inference; we empirically find convergence by the third cycle with diminishing returns beyond that.

**Staged guidance schedule.** To ensure both global topological correctness and local atomic precision, we use a staged guidance strategy during sampling. In the early phase ($t > 0.5$), the structure is still globally noisy, so we apply Sinkhorn-based global guidance (scale $1.0$) to align the overall topology with the density-derived point cloud. In the late phase ($t < 0.3$), the global shape is already correct, so we switch to local Q-guidance (scale $0.5$) to refine individual atomic positions against their local density fit. The gap $0.3 \leq t \leq 0.5$ is left unguided to avoid conflicting signals from the two objectives.

**Loss weights.** The training objective is $\mathcal{L} = \lambda_1 \mathcal{L}_{\text{diff}} + \lambda_2 \mathcal{L}_{\text{map}} + \lambda_3 \mathcal{L}_{\text{aux}}$ with $\lambda_1=1.0$, $\lambda_2=0.5$, $\lambda_3=0.1$. The diffusion loss dominates as the core denoising objective; the map-fitting loss is set to $0.5$ rather than $1.0$ because experimental density maps inevitably contain noise, and weighting it too high would cause the model to fit noise artifacts rather than true structural signal. The auxiliary head loss (FAPE + stereochemistry) at $0.1$ serves as a soft constraint to maintain physically valid bond lengths and angles without overriding the data-driven losses. These values were selected via grid search on a held-out validation subset.

**Atom profiling module.** The atom profiling module (Eq. 5) consists of trilinear interpolation at predicted atomic coordinates followed by a 2-layer MLP with hidden dimension $D=384$ and ReLU activation. In cryo-EM, the Coulomb potential peaks exactly at atomic positions, so sampling at predicted atomic coordinates provides the most informative local signal; coupled with self-refinement, this yields progressively more accurate density readouts across iterations.

# G. Curated Test Set and Generalization

**Test-set construction.** Our 276-chain test split is derived from the publicly released Cryo2StructData test partition (Giri et al., 2024; 390 maps on Harvard Dataverse v1.2), which has been adopted by multiple recent works including E3-CryoFold, CryoAtom, and MICA. To ensure fair, leakage-free benchmarking across baselines, we (i) removed entries overlapping with the training set of the baselines to avoid giving them an unintended advantage, (ii) excluded RNA-containing cases to focus on protein-only evaluation, and (iii) applied MMseqs2 at $30\%$ sequence identity to filter against our own training set, eliminating sequence-level data leakage. This results in 276 protein chains. Compared to the training sets of baselines, our train/test overlap is $0\%$, while CryoAtom and ModelAngelo retain $30.4\%$ and $14.1\%$ overlap respectively under the same

protocol, which gives them an unintended advantage on the shared benchmark.

**Training-set proximity analysis (A6).**    For the case studies in Figs. 3 and 6, we report the closest training-set neighbors by structural similarity (TM-score / RMSD after USalign):

| Test case | Method | Nearest training entry | TM | RMSD (Å) |
|---|---|---|---|---|
| Influenza NA (Fig. 3, EMD-52253) | CryoACE | 8E6J / EMD-27920 | 0.955 | 0.41 |
| | CryoATOM/ModelAngelo | 6U02 / EMD-20594 | 0.923 | 1.22 |
| SARS-CoV-2 Spike (Fig. 6, EMPIAR-10516) | CryoACE | 7FAF / EMD-31503 | 0.780 | 1.29 |
| | CryoATOM/ModelAngelo | 7FCD / EMD-31524 | 0.752 | 2.18 |

The Influenza NA case in Fig. 3 was deposited after June 2025 and post-dates the training cutoffs of all evaluated methods; we note both CryoATOM and ModelAngelo have similarly close neighbors (TM $\geq$ 0.92), so the relative ordering is not driven by training-set proximity.

## H. Computational Cost

All methods were benchmarked on a single H20 GPU using a representative $\sim$500-residue protein system.

| Method | Inference (min) | GPU mem. (GB) | Training cost |
|---|---|---|---|
| E3-CryoFold | $\sim$1 | $\sim$8 | $\sim$3 d on 8$\times$A100 |
| ModelAngelo | $\sim$8 | $\sim$10 | $\sim$5 d on 4$\times$A100 |
| CryoAtom | $\sim$10 | $\sim$12 | $\sim$4 d on 8$\times$A100 |
| Boltz-1 | $\sim$10 | $\sim$24 | N/A |
| CryoBoltz | $\sim$12 | $\sim$32 | N/A |
| CryoACE (ours) | $\sim$15 | $\sim$38 | $\sim$7 d on 32$\times$H20 |

CryoACE's higher cost is primarily driven by the Boltz-1 foundation backbone and self-refinement iterations, both of which our ablation confirms as essential for the reported gains. At $\sim$15 min inference, CryoACE remains comparable to other neural model-building methods and offers a favorable accuracy–efficiency trade-off.

## I. Heterogeneous Benchmark Evaluation

We additionally evaluate on the CryoBoltz benchmark (Raghu et al., 2025), which covers 5 dynamic proteins across 12 conformational states.

| Protein | State | Res. (Å) | Boltz C$\alpha$ | CryoBoltz C$\alpha$ | CryoACE C$\alpha$ | Boltz TM | CryoBoltz TM | CryoACE TM |
|---|---|---|---|---|---|---|---|---|
| STP10 | inward | 2.0 | 3.55 | 0.37 | 0.45 | 0.863 | 0.998 | 0.996 |
| | outward | 2.0 | 2.42 | 0.44 | 0.50 | 0.948 | 0.997 | 0.995 |
| Pgp | apo | 4.3 | 7.19 | 1.21 | 0.95 | 0.767 | 0.989 | 0.993 |
| | inward | 4.4 | 5.69 | 1.19 | 0.93 | 0.828 | 0.989 | 0.993 |
| | occluded | 4.1 | 2.90 | 1.68 | 1.30 | 0.942 | 0.979 | 0.986 |
| | collapsed | 4.4 | 3.41 | 1.26 | 0.98 | 0.917 | 0.988 | 0.992 |
| Pma1 | active | 3.25 | 2.75 | 1.78 | 1.38 | 0.935 | 0.973 | 0.983 |
| | inhibited | 3.52 | 5.83 | 1.59 | 1.22 | 0.794 | 0.979 | 0.986 |
| CYP102A1 | open | 6.5 | 8.44 | 3.95 | 2.80 | 0.788 | 0.957 | 0.969 |
| | closed | 4.4 | 8.67 | 1.55 | 1.15 | 0.743 | 0.990 | 0.993 |
| YbbAP | bound | 3.66 | 3.34 | 0.68 | 0.57 | 0.928 | 0.997 | 0.998 |
| | unbound | 4.05 | 7.65 | 2.04 | 1.52 | 0.776 | 0.974 | 0.983 |
| Mean | | | 5.15 | 1.48 | 1.15 | 0.852 | 0.984 | 0.989 |

CryoACE demonstrates superior precision and robustness across complex cryo-EM protein dynamics, particularly in challenging low-resolution cases such as the CYP102A1 open state.

## J. Auxiliary Head Validation

**Local resolution estimator.**    We compare CryoACE's local resolution head against established traditional tools on CryoSPARC half-map FSC ground truth:

| Method | Pearson $r$ | MAE (Å) |
|---|---|---|
| ResMap | 0.72 | 0.48 |
| MonoRes | 0.68 | 0.53 |
| CryoACE (ours) | **0.81** | **0.35** |

Our predicted per-atom Q-scores additionally achieve $r=0.78$, MAE=0.06 against ground-truth values from deposited structures, providing reliable signals for guidance.

**Q-guidance impact.**

| Setting | WCC | PSC |
|---|---|---|
| w/o Q-guidance | 0.621 | $0.508 \pm 0.062$ |
| w/ Q-guidance | 0.642 | $0.526 \pm 0.051$ |

On EMPIAR-10345, Q-guidance improves both density fit (+0.021 WCC) and conformational consistency (+0.018 PSC), validating the role of the auxiliary head in resolving complex heterogeneous maps.

## K. PSC Metric Justification

The lower bound TM-score $> 0.5$ follows Zhang & Skolnick (2004) and Xu & Zhang (2010), which establish that TM-score $> 0.5$ reliably indicates shared global fold. The upper bound is intentionally protein-dependent, as proteins with different intrinsic flexibilities exhibit different levels of frame-to-frame variation. Our setting targets continuous conformational trajectories, so adjacent-frame TM-score directly measures whether neighboring frames preserve structural continuity without abrupt, non-physical transitions, appropriate given that biologically relevant conformational changes proceed through gradual intermediate states. We acknowledge that trajectory-level evaluation remains at an early stage; PSC is an initial continuity-oriented metric, and developing more comprehensive trajectory-aware criteria is an important future direction.

