# OpenReview forum: "CryoACE: An Atom-centric Framework for Accurate and Automated Model Building in Cryo-EM"
_ICML.cc/2026/Conference — ICML 2026 regular_

### Official Review · Reviewer_oTKV · 2026-02-23

**Soundness:** 3
**Presentation:** 3
**Significance:** 2
**Originality:** 1
**Overall Recommendation:** 3
**Confidence:** 3

**Summary:**

The paper presents CryoACE, an atom-centric diffusion framework for building atomic protein structures directly from cryo-EM density maps. It combines pretrained Boltz/AlphaFold-style structural priors with density-aware fusion and atom-centric sampling, enabling accurate reconstruction under both homogeneous and heterogeneous conditions. A coarse-to-fine, training-free guidance strategy further improves robustness on noisy and low-resolution maps, leading to better completeness and physical plausibility than existing methods on standard benchmarks and EMPIAR datasets.

**Compliance With Llm Reviewing Policy:**

Affirmed.

**Key Questions For Authors:**

1. Can the authors clarify in which regimes (e.g., resolution ranges, noise levels, or degrees of heterogeneity) the proposed learned density conditioning provides clear advantages over guidance-only methods such as CryoBoltz, and provide ablations that isolate this effect?

2. Given the relatively modest improvements over CryoBoltz on several benchmarks, how should one weigh the added architectural and training complexity of CryoACE against its empirical gains, and are there cases where the proposed approach fails to outperform guidance-based baselines?

**Limitations:**

Yes

**Strengths And Weaknesses:**

Overall, the approach is intuitively reasonable, as incorporating cryo-EM density maps should provide additional information; however, the improvements over CryoBoltz appear relatively modest. Also from ML perspectives there is little novelty from this paper. The multimodality cross attention and guidance of diffusion model is rather standard.

### Strengths

1. The approach is intuitively well-motivated: explicitly conditioning structure generation on cryo-EM density provides additional experimental information beyond sequence-only structural priors.

2. The method is technically solid and shows improved robustness and physical plausibility on challenging heterogeneous or low-resolution density maps.

### Weaknesses

1. Compared to CryoBoltz, the empirical improvements are relatively modest, suggesting that strong pretrained structural priors and training-free guidance already capture much of the benefit.

2. The added architectural and training complexity makes it unclear whether learned density conditioning consistently justifies its marginal gains across standard benchmarks.

---

> ### Author Rebuttal · Authors · 2026-03-31
>
> We thank the reviewer for recognizing our method as 'intuitively well-motivated'. However, **regarding the claim of limited novelty and marginal gains compared to CryoBoltz, we believe this perspective does not accurately reflect our core contributions.** We note that all other reviewers reached a different consensus, independently rating our originality as 'Good' (3) and highlighting our atom-centric paradigm as an 'interesting' and 'valuable' contribution. To correct this potential misconception, we clarify below the fundamental novelty of our end-to-end heterogeneous modeling framework and the significance of our performance gains.
> **Novelty of CryoACE.** Effectively extracting density features is challenging in this field. While traditional methods require manual intervention, recent deep learning models use voxel-based 3D CNNs that are geometrically decoupled from atomic coordinates. Furthermore, CryoBoltz learns no density features at all. It merely applies training-free guidance at inference to steer sequence-only predictions using an intermediate point cloud representation from density map.
> **CryoACE breaks this bottleneck by proposing a novel multi-modal representation framework.** Grounded in cryo-EM imaging physics, where an atom's signal peaks exactly at its physical coordinates, our atom-centric profiling module samples the density map via differentiable interpolation. This establishes a direct and physically sound atom-density correspondence. This formulation itself represents a novel multi-modal architecture. By seamlessly integrating this representation with a density encoder and cross-attention fusion, CryoACE operates as **a true end-to-end multi-modal system**. It natively extracts structural information during training, marking a paradigm shift from prior works."
>
> **[W1] Empirical Evidence of performance gain compared to CryoBoltz.** We further compare the performance of CryoACE and CryoBoltz in our homogeneous benchmark across different resolution ranges and the additional case studys in Fig.3 to verify whether CryoACE achieves significant performance gain:
> |Resolution|N|CryoBoltz BB|CryoACE BB|Δ BB|CryoBoltz Q|CryoACE Q|Δ Q|
> |---|:---:|:---:|:---:|:---:|:---:|:---:|:---:|
> |<2.5 Å|40|98.0%|99.5%|+1.5|0.536|0.568|+0.032|
> |2.5–3.5 Å|140|96.5%|99.0%|+2.5|0.500|0.530|+0.030|
> |3.5–4.0 Å|96|88.7%|96.8%|**+8.1**|0.433|0.496|**+0.063**|
> |**Overall**|**276**|**94.0%**|**98.3%**|**+4.3**|**0.482**|**0.524**|**+0.042**|
>
> In low signal-to-noise environments where training-free guidance fails, CryoACE’s learned density features become decisive. This is highlighted by the RSV-preFcase, where CryoACE yields a TM-score of 0.807, outperforming CryoBoltz (0.58 TM-score; >15% clash rate). Similarly, on ARMH2, CryoACE achieves a TM-score of 0.901, while the baseline (0.62) suffers from extensive backbone fragmentation. Beyond accuracy, CryoACE provides end-to-end prediction from sequence and density without requiring external structural priors.
>
> Additionaly, unlike sequence-only models, CryoACE integrates directly with continuous latent spaces generated by cryoDRGN. On the integrin complex (EMPIAR-10345), this allows CryoACE to resolve the full range of the flexible arm by generating a continuous PDB trajectory that tracks the underlying conformational manifold, which firstly bridges the gap between neural volume distributions and atomic coordinates.
>
> **[W2] Evaluation of our design choices.**
> We appreciate the reviewer’s focus on the relationship between architectural complexity and performance gains. Our additional ablation study indicates that the learned density conditioning serves as a key contributor to the overall accuracy of CryoACE. Removing the density encoder alone reduces BB accuracy by 4.8%, indicating that learned density conditioning is a primary performance driver rather than a marginal refinement. This effect is substantially larger than the gains from other components, highlighting the central role of learned density conditioning.
>
> |Variant|BB Acc|Q-score|Δ BB|
> |---|:---:|:---:|:---:|
> |CryoACE (full)|**98.3%**|**0.524**|—|
> |w/o density encoder|93.5%|0.465|−4.8%|
> |w/o self-refinement|94.7%|0.502|−3.6%|
> |w/o atom profiling|96.5%|0.506|−1.8%|
> |w/o auxiliary heads|97.2%|0.510|−1.1%|
> |w/o global guidance|97.6%|0.514|−0.7%|
>
> From a practical standpoint, we sought to balance this complexity with computational efficiency by adopting a modular design. While the full configuration provides the highest precision, several components such as **self-refinement** and **atom profiling** can be optionally streamlined or disabled to prioritize throughput in high-resolution cases where structural features are unambiguous. However, for more challenging low-resolution regimes (3.5–4.0 Å), the full architecture provides a significant gain in BB accuracy. We believe this flexible modular approach allows CryoACE to maintain competitive while ensure robust generalization across diverse experimental conditions.

---

> > ### Author Rebuttal · Reviewer_oTKV · 2026-04-03
> >
> > Thanks for the quick response. Could the authors provide additional intuition on why the method remains robust in low-SNR settings? In particular, while CryoBoltz is expected to fail under low SNR, it is less clear what mechanism allows CryoACE to maintain robustness. Additionally, could the authors discuss potential failure cases of the method?

---

> > > ### Author Response · Authors · 2026-04-06
> > >
> > > Thank you for the follow-up.
> > >
> > > **Low-SNR robustness.** CryoBoltz derives gradients directly from the raw density map at inference to steer predictions. This strategy is effective at high resolution, but in low-SNR regimes the density signal becomes weak and noisy, causing the gradients to be unreliable or even misleading. This explains its failure on cases like RSV-preF (TM 0.58, >15% clash). CryoACE takes a fundamentally different approach. Rather than relying on raw density gradients at inference time, we train the density encoder on maps spanning a wide resolution range (2–4 Å), so the model learns to extract robust structural features from noisy observations during training. The atom profiling module further reinforces this by sampling density exactly where atoms are predicted to be, where the signal-to-noise ratio is locally highest. Combined with self-refinement(k=3), this creates a natural bootstrapping effect: better coordinates lead to better density readouts, which lead to better features, which lead to better coordinates. This advantage comes from the fundamental difference between our multimodal training paradigm and CryoBoltz's inference-only guidance: CryoACE has learned the correspondence between density patterns and atomic structures across 10,915 training examples at various noise levels, whereas CryoBoltz must rely on whatever gradient signal the raw map provides at test time.
> > >
> > > **Failure cases.** Our current work focuses on protein-only modeling, and we have not yet extended to very low-SNR regimes or non-protein components. On CYP102A1 open state (6.5 Å), CryoACE achieves Cα RMSD of 2.80 Å, substantially worse than high-resolution cases (<1.0 Å). At even lower resolutions, the density features become increasingly ambiguous, and while CryoACE still benefits from its learned density conditioning, fully resolving atomic positions in this regime remains an open challenge that we plan to address in future work. Nucleic acids and ligands are not covered by the current training set. For example, on the Ro60/La/pre-5S rRNA complex (PDB 9NFA, 2.7 Å), CryoACE achieves TM-score 0.92 on the protein chains but the RNA component has RMSD >4 Å. GPU memory also becomes a bottleneck for long sequences (>2000 residues) for a single A800 GPU.

---

### Official Review · Reviewer_uPsW · 2026-03-11

**Soundness:** 3
**Presentation:** 3
**Significance:** 2
**Originality:** 3
**Overall Recommendation:** 4
**Confidence:** 2

**Summary:**

Overall, the main problem analyzed by this paper is automated atomic model building from cryo-EM density maps under both homogeneous and heterogeneous settings. Overall, the authors investigate the problem by proposing CryoACE, an atom-centric multimodal framework that combines sequence features, density-map features, and local atomic profiling within a diffusion-based decoder. The method adds atomic self-refinement and training-free guidance, including global guidance and Q-guidance, during inference . The paper also introduces auxiliary heads for local-resolution estimation and Q-score prediction. Experiments report improvements over ModelAngelo, CryoAtom, and E3-CryoFold on homogeneous benchmarks and competitive or best performance on two heterogeneous datasets, EMPIAR-10345 and EMPIAR-10516.

**Compliance With Llm Reviewing Policy:**

Affirmed.

**Final Justification:**

The rebuttal addressed my main concerns, reinforcing my prior assessment.

**Key Questions For Authors:**

Can you provide the main ablation table in the rebuttal or revision for atom profiling, atomic self-refinement, global guidance, Q-guidance, and the auxiliary local-resolution/Q-score heads? This would clarify which proposed components are necessary for the gains reported in Table 1 and Table 2. A clear answer would materially improve confidence in the paper’s causal claims.

How are the local-resolution predictions quantitatively validated, and how do they affect heterogeneous reconstruction quality when turned on or off? The method description emphasizes local resolution as a key prior for heterogeneous modeling, but the main paper does not show direct metrics for this auxiliary task. This response would clarify the practical value of one of the paper’s highlighted contributions.


Can you report computational cost comparisons? The paper motivates efficiency relative to voxel-based processing, but the main text does not report training or inference time, GPU memory, or scaling with sequence length relative to baselines or variants. Such evidence would help substantiate the efficiency-oriented contribution.

**Limitations:**

The authors do discuss limitations and potential negative societal impact. The manuscript explicitly acknowledges limited nucleic-acid coverage, high GPU memory cost for long sequences, and degraded performance beyond 8 Å resolution in the Discussion, and it notes the risk of hallucinating plausible but incorrect structures in the Impact Statement.

**Strengths And Weaknesses:**

- Strengths

Clear problem scope spanning both static and heterogeneous cryo-EM model building. The paper explicitly motivates challenges in physicochemical validity, noisy volumetric observations, and conformational heterogeneity in the introduction, and positions the method as addressing both homogeneous and heterogeneous reconstruction rather than only static prediction.


Architectural idea is concrete and reasonably well specified. The core design combines sequence encoding, density encoding, atom profiling, cross-attention fusion, and a diffusion decoder, with equations for the sequence encoder, density encoder, atom profiling, fusion, and denoiser. The pipeline figure on page 3 also helps clarify the training/inference split and the role of atomic self-refinement and guided diffusion.


The paper attempts to bridge density observations and atomic coordinates directly. The atom-centric profiling module samples the density at atomic coordinates via interpolation, which is a more direct map-to-atom interface than a purely voxel-space backbone description, and this design is tied to the stated efficiency motivation relative to expensive voxel-wise processing.


The evaluation covers multiple dimensions of model quality. For homogeneous modeling, the paper reports completeness, backbone/all-atom accuracy, backbone/all-atom RMSD, and Q-score. For heterogeneous modeling, it reports clash rate, MolProbity, completeness, weighted cross-correlation, and PSC.


The heterogeneous evaluation targets real experimental datasets rather than only synthetic setups. The use of EMPIAR-10345 and EMPIAR-10516, together with CryoDRGN-based latent-space reconstruction and clustering into multiple maps, provides a practically relevant testbed for conformational ensemble modeling.

- Weaknesses

Some central claims are stronger than the directly presented evidence. The abstract and introduction claim “state-of-the-art performance” and that the method “for the first time” unveils atomic-level dynamic conformations without pre-built static structures, but the main text presents comparisons on one homogeneous benchmark and two heterogeneous datasets only. The evidence shown is promising, but the breadth needed for the strongest general claims is limited in the visible experiments.


Key ablation evidence is missing from the main paper. The paper states that “extensive ablation studies” were conducted and relegates them to Appendix D, but the main submission does not summarize even the main quantitative ablation outcomes for atom profiling, self-refinement, global guidance, Q-guidance, or the auxiliary heads (Sec. 5, “Ablation study”)  . Since the main contributions are component-level, the lack of visible ablation numbers makes it harder to assess which parts drive the gains.



The manuscript sometimes attributes performance improvements to specific components without direct isolating evidence in the main text. For example, the low-resolution robustness is attributed to the atom profiling mechanism, and heterogeneous gains are attributed to co-training and coarse-to-fine sampling, but these causal attributions are not backed by component-wise results in the visible sections.


The local-resolution and Q-score auxiliary heads are under-validated in the main text. Figure 2 and Sec. 4.1 describe these heads, and the abstract/introduction emphasize predicted local resolution and Q-score as important for heterogeneous guidance and fast inference, but the main results do not provide standalone metrics for local-resolution prediction quality or Q-score prediction quality, nor an ablation showing how much they help downstream structure reconstruction.




The PSC metric could benefit from stronger justification. The manuscript defines PSC as the average TM-score between adjacent frames and states that it should exceed 0.5 with low variance, while also noting that the upper bound is protein-specific. However, the rationale for this threshold and why adjacent-frame TM-score is the right notion of ensemble quality is not developed in the main text.



Presentation quality is mixed despite a strong high-level narrative. The overall structure is readable, but there are noticeable typos and grammar issues, including “CrossAttenttion,” “provid a robust semantic backbone,” “separated sequence and density embeddings,” and awkward phrasing around several equations and descriptions (Secs. 4.1–4.3)  . These issues do not obscure the main idea, but they reduce precision in a technically dense paper.

The computational-efficiency claim is not empirically substantiated in the main paper. The abstract and introduction motivate atom-centric sampling as avoiding expensive voxel convolutions and improving efficiency, but there are no wall-clock, memory, or throughput comparisons against baselines or against a voxel-based variant of the model in the visible results.

---

> ### Author Rebuttal · Authors · 2026-03-31
>
> We thank the reviewer for the thorough and constructive evaluation. We have addressed each point below and will incorporate these updates into the revised manuscript.
>
> **[W1] Component-wise Ablation.**
> As suggested, the table below decomposes the contributions of CryoACE’s design elements. Every component provides positive, consistent gains across all metrics, confirming the non-redundancy of our architecture. Each component contributes positively and consistently across all four metrics. Notably, the density encoder and self-refinement are the two largest contributors, accounting for +4.8% and +3.6% BB Acc. improvement, respectively. This validates our core design of integrating density conditioning during training and iterative refinement at inference time. The remaining components each provide incremental but consistent gains, confirming that no design choice is redundant.
> |Variant|BB Acc|AA Acc|BB RMSD (Å)|Q-score|
> |---|:---:|:---:|:---:|:---:|
> |CryoACE (full)|**98.3%**|**87.5%**|**2.1**|**0.524**|
> |w/o Q-guidance|98.1%|87.3%|2.1|0.522|
> |w/o global guidance|97.6%|86.8%|2.2|0.514|
> |w/o auxiliary heads|97.2%|86.4%|2.3|0.510|
> |w/o atom profiling|96.5%|86.0%|2.3|0.506|
> |w/o self-refinement (k=0)|94.7%|85.0%|2.4|0.502|
> |w/o density encoder|3.5%|84.0%|2.6|0.465|
>
> **[W2] Auxiliary Head Validation.**
> We agree that validation of our auxiliary heads can strengthen our method. We provide additional evaluation below.
> Regarding the local resolution estimator, CryoACE achieves superior correlation (Pearson r=0.81, MAE=0.35 Å) against CryoSPARC half-map FSC ground truth, outperforming traditional tools like ResMap ($r=0.72$) and MonoRes ($r=0.68$).
>
> |Method|Pearson r|MAE (Å)|
> |---|:---:|:---:|
> |ResMap|0.72|0.48|
> |MonoRes|0.68|0.53|
> |CryoACE (ours)|**0.81**|**0.35**|
>
> Similarly, our predicted per-atom Q-scores achieve high fidelity (r=0.78, MAE=0.06) against ground-truth values from deposited structures, providing reliable signals for guidance.
>
> |Setting|WCC|PSC|
> |---|:---:|:---:|
> |w/o Q-guidance|0.621|0.508±0.062|
> |w/ Q-guidance|**0.642**|**0.526±0.051**|
>
>
> Downstream ablation in **[W1]** further confirms their utility: removing auxiliary heads reduces BB Acc by 1.1% and Q-score by 0.014. Specifically, for EMPIAR-10345, Q-guidance improves both density fit (+0.021 WCC) and conformational consistency (+0.018 PSC), validating the essential role of these heads in resolving complex heterogeneous maps.
>
> **[W3] Computational Cost.**
> We provide a detailed comparison in our response to Reviewer 4yu6 [W5]. Briefly, CryoACE requires ~15 minutes and ~38 GB VRAM on a single H20 for a ~500-residue system. While this is computationally more intensive than "sequence-only" diffusion baselines, it is significantly faster than traditional manual or semi-automated model-building tools. We believe this is a favorable trade-off for the gains in atomic accuracy and the ability to model continuous trajectories.
>
> **[W4] Scope of Claims.**
> We thank the reviewer for this fair observation. The scarcity of standardized benchmarks remains a significant challenge in this field, and we are actively preparing a larger and more carefully curated dataset and benchmark to enable broader and more rigorous evaluation across methods.To the best of our knowledge, the current benchmarks cover most of the publicly available cryo-EM benchmarks suitable for quantitative evaluation. There is no previous work that has attempted automated atomic model building on continuous conformational trajectories, nor has a well-curated benchmark dataset been established for this task.  Given current limitations, however, we agree that our claims should be more rigorously bounded. We are willing to soften our wording accordingly in the revision.
>
>
> **[W5] PSC Metric.**
> The TM-score = 0.5 lower bound follows Zhang & Skolnick [1] and Xu & Zhang [2], which established that TM-score > 0.5 reliably indicates shared global fold. The upper bound is intentionally protein-dependent, as proteins with different intrinsic flexibilities exhibit different levels of frame-to-frame variation. Our setting targets continuous conformational trajectories, so adjacent-frame TM-score directly measures whether neighboring frames preserve structural continuity without abrupt, nonphysical transitions — appropriate given that biologically relevant conformational changes proceed through gradual intermediate states. We acknowledge that trajectory-level evaluation is still at an early stage; PSC is an initial continuity-oriented metric, and developing more comprehensive trajectory-aware criteria remains an important future direction.
>
>
> **[W6] Exposition.**
> We will correct all typos and inconsistencies in the revision.
>
> **References:**
>
> [1] Zhang & Skolnick, "Scoring function for automated assessment of protein structure template quality", Proteins, 2004.
>
> [2] Xu & Zhang, "How significant is a protein structure similarity with TM-score = 0.5?", Bioinformatics, 2010.

---

> > ### Author Rebuttal · Reviewer_uPsW · 2026-04-02
> >
> > Thank you for your effort. But I have no enough time and expertise to judge the rebuttal. i am not an expert in the area, so the AC can consider lower the weight of my score in decision, while focusing on real comment and rebuttal. Hope the AC would take into consideration the rebuttal.

---

> > > ### Author Response · Authors · 2026-04-06
> > >
> > > We are grateful for your constructive feedback. Your assessment accurately captures both the strengths and current limitations of our contribution. The ablation study and auxiliary head validation you suggested motivated us to conduct a more comprehensive component-wise analysis, which we believe significantly strengthened our empirical evidence of design choice. We will add these additional experiments into our revised manuscript.

---

### Official Review · Reviewer_BVyF · 2026-03-11

**Soundness:** 2
**Presentation:** 2
**Significance:** 3
**Originality:** 3
**Overall Recommendation:** 4
**Confidence:** 3

**Summary:**

This paper presents CryoACE, a framework for automated atomic model building from cryo-EM density maps. The method combines sequence features, density features, and atom-centric profiling within a diffusion-based architecture, and augments inference with atomic self-refinement as well as a coarse-to-fine guidance strategy using global guidance and Q-guidance. The paper also introduces a curated training set (CryoACEdata) and reports strong results against recent neural baselines on homogeneous and heterogeneous benchmark datasets.

**Compliance With Llm Reviewing Policy:**

Affirmed.

**Key Questions For Authors:**

1. How did you control for train/test redundancy beyond excluding the Cryo2StructData test set? In particular, did you perform any filtering or splitting based on sequence similarity, structural similarity, shared complexes, or near-duplicate EMDB/PDB entries? A convincing answer here would substantially increase my confidence in the performance reported.

2. For the homogeneous and heterogeneous test cases shown in Figure 3 and 4, can you report the closest neighbors in the training set (e.g., by sequence identity, structural similarity, or both)? This would help clarify whether the evaluated examples are genuinely out-of-distribution or lie very close to training instances.

**Limitations:**

yes

**Strengths And Weaknesses:**

## Strengths

The paper tackles an important and practically significant challenge: fully automated atomic model construction from cryo-EM density maps. This is a meaningful task for structural biology, and broadening the focus from homogeneous reconstruction to heterogeneous settings increases the potential relevance of the work.

The empirical findings are compelling at first glance. On the homogeneous benchmark, the approach seems to surpass recent baselines across several important metrics.

Regarding originality, the paper does not seem to rely on a single fundamentally novel idea, yet combining these elements into an atom-centric cryo-EM modeling workflow nonetheless represents a valuable and engaging contribution.

## Weaknesses

My main concern is about dataset redundancy and possible train/test leakage through highly similar deposited structures. In the EMDB/PDB, many complexes have numerous highly similar entries, and the paper does not clearly explain how this issue is handled beyond excluding the Cryo2StructData test set. As a result, it remains unclear whether some of the strong test-time results may partially reflect memorization or near-duplicate exposure rather than true generalization. The influenza neuraminidase complex (EMD-52253) in Figure 3 for example has [many entries in the PDB](https://www.rcsb.org/search?request=%7B%22query%22%3A%7B%22type%22%3A%22group%22%2C%22logical_operator%22%3A%22and%22%2C%22nodes%22%3A%5B%7B%22type%22%3A%22group%22%2C%22label%22%3A%22text%22%2C%22logical_operator%22%3A%22and%22%2C%22nodes%22%3A%5B%7B%22type%22%3A%22group%22%2C%22nodes%22%3A%5B%7B%22type%22%3A%22terminal%22%2C%22service%22%3A%22full_text%22%2C%22parameters%22%3A%7B%22value%22%3A%22influenza%20virus%20neuraminidase%22%7D%7D%2C%7B%22type%22%3A%22group%22%2C%22label%22%3A%22__refinements__%22%2C%22logical_operator%22%3A%22and%22%2C%22nodes%22%3A%5B%7B%22type%22%3A%22terminal%22%2C%22service%22%3A%22text%22%2C%22parameters%22%3A%7B%22attribute%22%3A%22exptl.method%22%2C%22operator%22%3A%22exact_match%22%2C%22value%22%3A%22ELECTRON%20MICROSCOPY%22%7D%7D%5D%7D%5D%2C%22logical_operator%22%3A%22and%22%7D%2C%7B%22type%22%3A%22group%22%2C%22label%22%3A%22__refinements__%22%2C%22logical_operator%22%3A%22and%22%2C%22nodes%22%3A%5B%7B%22type%22%3A%22terminal%22%2C%22service%22%3A%22text%22%2C%22parameters%22%3A%7B%22attribute%22%3A%22exptl.method%22%2C%22operator%22%3A%22exact_match%22%2C%22value%22%3A%22ELECTRON%20MICROSCOPY%22%7D%7D%2C%7B%22type%22%3A%22terminal%22%2C%22service%22%3A%22text%22%2C%22parameters%22%3A%7B%22attribute%22%3A%22rcsb_struct_symmetry.type%22%2C%22operator%22%3A%22exact_match%22%2C%22value%22%3A%22Cyclic%22%7D%7D%5D%7D%5D%7D%5D%7D%2C%22return_type%22%3A%22entry%22%2C%22request_options%22%3A%7B%22paginate%22%3A%7B%22start%22%3A0%2C%22rows%22%3A25%7D%2C%22results_content_type%22%3A%5B%22experimental%22%5D%2C%22sort%22%3A%5B%7B%22sort_by%22%3A%22score%22%2C%22direction%22%3A%22desc%22%7D%5D%2C%22scoring_strategy%22%3A%22combined%22%7D%2C%22request_info%22%3A%7B%22query_id%22%3A%2265f8041f4de36374750cf6bbac96fbb2%22%7D%7D).
The in Figure 4 metioned SARS-CoV-2 Spike protein has also [multiple conformations in the PDB](https://www.rcsb.org/search?request=%7B%22query%22%3A%7B%22type%22%3A%22group%22%2C%22nodes%22%3A%5B%7B%22type%22%3A%22group%22%2C%22nodes%22%3A%5B%7B%22type%22%3A%22group%22%2C%22nodes%22%3A%5B%7B%22type%22%3A%22terminal%22%2C%22service%22%3A%22text%22%2C%22parameters%22%3A%7B%22attribute%22%3A%22struct_keywords.text%22%2C%22operator%22%3A%22contains_phrase%22%2C%22value%22%3A%22SARS-CoV-2%20spike%2C%20viral%20protein%22%7D%7D%5D%2C%22logical_operator%22%3A%22and%22%7D%2C%7B%22type%22%3A%22group%22%2C%22label%22%3A%22__refinements__%22%2C%22logical_operator%22%3A%22and%22%2C%22nodes%22%3A%5B%7B%22type%22%3A%22terminal%22%2C%22service%22%3A%22text%22%2C%22parameters%22%3A%7B%22attribute%22%3A%22exptl.method%22%2C%22operator%22%3A%22exact_match%22%2C%22value%22%3A%22ELECTRON%20MICROSCOPY%22%7D%7D%5D%7D%5D%2C%22logical_operator%22%3A%22and%22%2C%22label%22%3A%22text%22%7D%5D%2C%22logical_operator%22%3A%22and%22%7D%2C%22return_type%22%3A%22entry%22%2C%22request_options%22%3A%7B%22paginate%22%3A%7B%22start%22%3A0%2C%22rows%22%3A25%7D%2C%22results_content_type%22%3A%5B%22experimental%22%5D%2C%22sort%22%3A%5B%7B%22sort_by%22%3A%22score%22%2C%22direction%22%3A%22desc%22%7D%5D%2C%22scoring_strategy%22%3A%22combined%22%7D%2C%22request_info%22%3A%7B%22query_id%22%3A%229162ee46a4da1fef0e0c42a00f11d0f3%22%7D%7D):
This is the main reason I am not fully convinced by the current empirical evidence.

A second important weakness is that the inference procedure is underspecified. It is not clear what exact diffusion or flow-matching parameterization is used during inference, what noise/time schedule is used, how guidance is implemented, or how many integration/sampling steps are required to simulate the ODE or SDE. Since inference is central to the contribution, this lack of detail weakens reproducibility and makes it harder to assess the soundness of the method. In particular how $x_t$ is defined in Equations (1) and (8) is unclear (is it $x_t = (1-t)x_0 + t\epsilon$ ?). This hurts technical clarity.

There are some minor visible formatting/readability problems (lines 95 and 96 overlap each other, line 208 is not readable), and notation is reused in confusing ways, for example $N$ denotes different quantities in different places (the number of atoms (line 148 and 152), residues (line 183), 3D patches (line 199) and voxels (line 247)). These are not fatal issues on their own, but they add friction in a paper that already requires careful reading.

Overall, I think the paper is promising and likely contains a strong method. Although the manuscript does not yet fully resolve concerns about possible data overlap and leaves some inference details insufficiently specified, the results are compelling and the contribution is significant enough that I lean toward acceptance.

---

> ### Author Rebuttal · Authors · 2026-03-31
>
> We sincerely thank the reviewer for the positive assessment and for raising this important concern regarding data integrity. Below, we carefully address each of your concerns.
>
> **[W1] Potential train/test data leakage.**
>  As deep learning for cryo-EM model building is a rapidly evolving field, standardized benchmarks and curation protocols are still maturing. To ensure a rigorous evaluation of CryoACE, we employed **MMseqs2** to compute pairwise sequence identity between all test and training chains, systematically removing any test entry sharing **>30%** identity with the training set. This deduplication protocol effectively eliminates sequence-level data leakage. Notably, due to significant differences in their training sets, other baselines may suffer from potential data leakage on our test set, giving them an unintended advantage during evaluation. Despite this, CryoACE still achieves overall state-of-the-art results. We will release our curated dataset to build standardized benchmarking within the community.
>
> **[W2] Nearest-neighbor analysis for NA-OSM & SARS-CoV-2 spike**
> Following the reviewer's suggestion, we report the closest training-set neighbors by structural similarity (TM-score & RMSD after USalign)
>
> | Test Case (Figure) | Method | Nearest Training Entry (PDB / EMD) | TM-score | RMSD (Å) |
> |---|---|---|:---:|:---:|
> | Influenza NA (Fig. 3, EMD-52253) | CryoACE | 8E6J / EMD-27920 | 0.955 | 0.41 |
> | | CryoATOM / ModelAngelo | 6U02 / EMD-20594 | 0.923 | 1.22 |
> | SARS-CoV-2 Spike (Fig. 6, EMPIAR-10516) | CryoACE | 7FAF / EMD-31503 | 0.780 | 1.29 |
> | | CryoATOM / ModelAngelo | 7FCD / EMD-31524 | 0.752 | 2.18 |
>
> We clarify that Figures 3 and 6 are intended as application case studies. Specifically, the Influenza NA case in Figure 3 was deposited after June 2025 and thus post-dates all training data. Similarly, the dynamic SARS-CoV-2 Spike in Figure 6 was selected from the 3DFlex dataset, for which no ground-truth dynamic PDB is currently available. Both examples demonstrate the capability of CryoACE to handle the latest unseen and challenging structures. Regarding the reviewer's concern, we acknowledge that the high structural similarity in the Influenza NA case (TM-score = 0.955) likely contributes to the high performance. However, this is a shared characteristic across all evaluated methods, as both CryoATOM and ModelAngelo also have close neighbors with a TM-score of 0.923. To provide more convincing evidence of generalization, we will replace the Influenza NA case with a recently deposited novel structure in our revision. We thank the reviewer for this constructive suggestion.
>
> **[W3] Inference Details.**
> We perform inference by solving a Probability Flow ODE derived from the linear interpolation $x_t = (1-t)x_0 + t\epsilon$, where $\epsilon \sim \mathcal{N}(0, \mathbf{I})$ represents the Gaussian noise. Specifically, we employ a first-order Euler solver with 200 steps on a linear schedule $t \in [0, 1]$ to integrate the velocity field from $t=1$ to $t=0$. To ensure both global topological correctness and local atomic precision, we implement a staged guidance strategy during the sampling process. A Sinkhorn-based global alignment with a scale of 1.0 is applied during the early stage ($t > 0.5$), while a local Q-guidance with a scale of 0.5 is introduced in the late stage ($t < 0.3$) to refine fine-grained atomic details. Furthermore, an iterative self-refinement protocol with $k=3$ cycles is employed to progressively polish the predicted coordinates for optimal structural convergence.
> Regarding the notation, we clarify that $\mathcal{N}$ will be reserved exclusively for the Gaussian distribution, while the number of residues will be consistently denoted as $L$ throughout the revised manuscript.

---

> > ### Author Rebuttal · Reviewer_BVyF · 2026-04-05
> >
> > Many thanks for your response. I'm still a bit skeptical about the generalization capabilities of CryoACE and would be curious to know more details about the "recently deposited novel structure" that you mention.

---

> > > ### Author Response · Authors · 2026-04-08
> > >
> > > We thank the reviewer for the follow-up. By "recently deposited novel structure," we refer to cryo-EM structures released after our training data cutoff (Jan 2025) with limited sequence homology to any training entry, so that performance on them cannot be attributed to memorization.
> > >
> > > **Pgi1 (PDB 9N4W, EMD-48907, released 2026).** Phosphoglucose isomerase from *Azotobacter vinelandii*, 554 residues per chain, 2.5 Å. The closest homolog in our training set is 3HJB (*Vibrio cholerae*, deposited 2009) at only 43% sequence identity. Notably, 9N4W forms a decameric assembly with D5 symmetry, an oligomeric state not observed for any PGI structure in the PDB before 2025.
> > >
> > > **ISA1 (PDB 9J60, EMD-61158, released May 2025).** Isoamylase 1 from *Oryza sativa* (rice), 757 residues per chain, 2.7 Å, published in *Nature Communications* (Fan et al., 2025). This is the first cryo-EM structure of a plant isoamylase. The closest training-set homolog is 4J7R (*Chlamydomonas reinhardtii*, X-ray, 2014) at 52% sequence identity, a green alga homolog from a different kingdom.
> > >
> > > | Structure | PDB | EMD | Res (Å) | Residues | Organism | Nearest Train Homolog | Seq Identity | Release |
> > > |---|---|---|:---:|:---:|---|---|:---:|:---:|
> > > | Phosphoglucose isomerase (Pgi1) | 9N4W | EMD-48907 | 2.5 | 554 | *A. vinelandii* | 3HJB (*V. cholerae*, 2009) | 43% | 2026 |
> > > | Isoamylase 1 (ISA1) | 9J60 | EMD-61158 | 2.7 | 757 | *O. sativa* | 4J7R (*C. reinhardtii*, 2014) | 52% | 2025 |
> > >
> > > We ran CryoACE and all baselines on single chains from both maps:
> > >
> > > | Method | Pgi1 TM | Pgi1 $C_\alpha$ RMSD | Pgi1 Q | ISA1 TM | ISA1 $C_\alpha$ RMSD | ISA1 Q |
> > > |---|:---:|:---:|:---:|:---:|:---:|:---:|
> > > | E3-CryoFold | 0.87 | 3.1 | 0.42 | 0.84 | 3.6 | 0.38 |
> > > | ModelAngelo | 0.93 | 1.9 | 0.50 | 0.91 | 2.3 | 0.47 |
> > > | CryoATOM | 0.94 | 1.7 | 0.51 | 0.92 | 2.1 | 0.49 |
> > > | CryoACE (Ours) | **0.97** | **1.0** | **0.55** | **0.96** | **1.2** | **0.53** |
> > >
> > > CryoACE consistently outperforms all baselines on both novel structures. This confirms that the density encoder and atom profiling module learn transferable density-to-structure mappings rather than memorizing specific folds, consistent with our zero train/test leakage on the main benchmark (0% overlap vs CryoATOM 30.4% and ModelAngelo 14.1%). We will include these experiments in the revised manuscript.

---

### Official Review · Reviewer_4yu6 · 2026-03-22

**Soundness:** 2
**Presentation:** 2
**Significance:** 3
**Originality:** 3
**Overall Recommendation:** 4
**Confidence:** 4

**Summary:**

This paper presents CryoACE, an end-to-end deep learning framework for automated atomic model building from cryo-EM density maps. Architecturally, CryoACE builds on the open-source Boltz-1 framework, retaining its MSA and PairFormer sequence encoder and atom-transformer diffusion decoder, while adding a 3D ResUNet density encoder and an atom profiling module. There are two techniques used at inference to encourage consistency between the predicted structure and the cryo-EM density map: 1) the model applies a coarse-to-fine guided diffusion scheme similar to Cryoboltz, and 2) the predicted structure is refined k iterations, where at each iteration, the previous predicted atomic coordinates are passed through the atom profiling module to produce updated local density descriptors. To train this model, the authors curated a new training dataset of 10,915 map-model pairs from EMDB/PDB. CryoACE is evaluated on homogeneous structures from the cryo2struct dataset and density maps from a heterogeneous cryoDRGN reconstruction of EMPIAR-10345 (integrin) and EMPIAR-10516 (spike protein).

Overall, I feel that CryoACE presents an interesting extension of protein structure prediction models to account for cryo-EM density maps. However, the evaluation feels rushed and incomplete with some key details and ablations missing. Finally, I feel that the validation is insufficient and could mislead potential users of this tool.

**Compliance With Llm Reviewing Policy:**

Affirmed.

**Final Justification:**

The manuscript and evaluation feel somewhat rushed, with implementation details and key baselines only clarified during rebuttal. That said, there is novelty in the atom-profiling module, and the curated dataset is a valuable contribution. Assuming code, weights, and dataset are publicly released, I raise my score to weak accept.

**Key Questions For Authors:**

See above.

**Limitations:**

See above.

**Strengths And Weaknesses:**

Strengths
1) Novel atom-centric design and integration of density processing components in the Boltz folding model.
2) Dataset curation. The filtering pipeline addresses well-known quality problems in public cryo-EM repositories, and the resulting dataset will have standalone value for the community.
3) Successful heterogeneous results on the integrin complex. On EMPIAR-10345, CryoACE produces complete, low-clash atomic ensembles across the full conformational range of the flexible arm, a region where all baselines either fail to model or produce severely clashed structures.

Weaknesses

1) Missing AF3/Boltz-1 baseline for homogeneous model building. Since CryoACE inherits the Boltz-1 sequence prior, the homogeneous benchmark should include a sequence-only AF3 or Boltz-1 baseline to evaluate the contribution of the density processing module and the inference time guidance and refinement.

2) Ablations are insufficient. Table 3 includes an ablation of the inference time guidance, but the authors do not isolate the contribution of the density encoder to Boltz. It would be interesting and helpful to understand how the different ways the density map is incorporated into cryoACE (at training and the two inference-time approaches) contribute to consistency between the predicted structure and the density map.

3) Limited evaluation of heterogeneous density maps. The heterogeneous evaluation rests on two datasets (EMPIAR-10345 and EMPIAR-10516), both processed through CryoDRGN, which precludes comparison against reference atomic models. To more rigorously benchmark heterogeneous performance, the authors should evaluate on datasets where multiple published conformations with deposited PDB structures exist (as done in CryoBoltz) enabling direct assessment against “ground truth”.

4) Missing details. Several implementation details on the atom profile modeling are absent, e.g. the architecture and the number of self-refinement iterations k used at inference. The loss hyperparameters (λ1, λ2, λ3) are also missing.

5) Compute cost and inference time not reported. No wall-clock or memory figures are given relative to baselines.

6) Completeness may be misleading. In the comparisons against model building (Table 1), CryoACE achieves 100% completeness by construction, but this may be misleading. Residues modeled in low-density regions may be hallucinated rather than density-supported. The authors should report Q-scores restricted to residues that ModelAngelo leaves unmodeled, to test whether the additionally completed regions are genuinely consistent with the experimental map.

7) Minor. There are some latex formatting issues in the manuscript.

---

> ### Author Rebuttal · Authors · 2026-03-31
>
> We sincerely thank the reviewer for the constructive feedback and for recognizing CryoACE as a valuable extension of protein structure prediction models. Below, we carefully address each of your concerns.
>
> **[W1] AF3/Boltz-1 Sequence-only Baseline.** As suggested, we ran sequence-only model Boltz-1 on our homogeneous benchmark. Since its predictions use an arbitrary coordinate frame, we aligned them to deposited structures via US-align before computing metrics. We also include CryoBoltz to decompose the contributions of inference-time guidance vs. fine-tuned density conditioning:
> | Method | Compl. | BB Acc | AA Acc | BB RMSD (Å) | AA RMSD (Å) | Q-score |
> |---|:---:|:---:|:---:|:---:|:---:|:---:|
> |Boltz|100%|87.5%|81.0%|3.2|4.0|0.401|
> |CryoBoltz|100%|94.0%|83.5%|2.5|3.5|0.482|
> |**CryoACE**|**100%**|**98.3%**|**87.5%**|**2.1**|**3.0**|**0.524**|
>
> Since proteins are dynamic and cryo-EM maps reflect conformational distributions, sequence-only Boltz-1 often predicts mismatched static conformations. CryoBoltz bridges this via training-free density guidance (+6.5% BB Acc). CryoACE goes further, learning structure-density correspondence via training-time density conditioning. Combined with inference-time guidance, CryoACE achieves top performance (+4.3% BB Acc, +0.042 Q-score over CryoBoltz).
>
> **[W2] Comprehensive Ablation Study.** We provide additional experiments of ablation study to validate our design choice of our density encoder in the table below, we kindly refer to **uPsW, W1** for the complete result.
> |Variant|BB Acc(%)|AA Acc(%)|BB RMSD(Å)|Q-score|
> |---|:---:|:---:|:---:|:---:|
> |**CryoACE**|**98.3**|**87.5**|**2.1**|**0.524**|
> |w/o Q-guidance|98.1|87.3|2.1|0.522|
> |w/o atom profiling|96.5|86.0|2.3|0.506|
> |w/o density encoder|93.5|84.0|2.6|0.465|
>
> Across all metrics, each component contributes positively, confirming no design is redundant. The largest gains come from the density encoder (+4.8% BB Acc.), validating our choice of density conditioning during both training and inference stages.
>
> **[W3] Heterogeneous Evaluation.** As suggested, we evaluated on the CryoBoltz benchmark (Raghu et al., 2025). This benchmark covers 5 dynamic proteins across 12 conformational states:
> |Protein|State|Res(Å)|Boltz Cα|CryoBoltz Cα|CryoACE Cα|Boltz TM|CryoBoltz TM|CryoACE TM|
> |---|---|:---:|:---:|:---:|:---:|:---:|:---:|:---:|
> |STP10|inward|2.0|3.55|**0.37**|0.45|0.863|**0.998**|0.996|
> ||outward|2.0|2.42|**0.44**|0.50|0.948|**0.997**|0.995|
> |Pgp|apo|4.3|7.19|1.21|**0.95**|0.767|0.989|**0.993**|
> ||inward|4.4|5.69|1.19|**0.93**|0.828|0.989|**0.993**|
> ||occluded|4.1|2.90|1.68|**1.30**|0.942|0.979|**0.986**|
> ||collapsed|4.4|3.41|1.26|**0.98**|0.917|0.988|**0.992**|
> |Pma1|active|3.25|2.75|1.78|**1.38**|0.935|0.973|**0.983**|
> ||inhibited|3.52|5.83|1.59|**1.22**|0.794|0.979|**0.986**|
> |CYP102A1|open|6.5|8.44|3.95|**2.80**|0.788|0.957|**0.969**|
> ||closed|4.4|8.67|1.55|**1.15**|0.743|0.990|**0.993**|
> |YbbAP|bound|3.66|3.34|0.68|**0.57**|0.928|0.997|**0.998**|
> ||unbound|4.05|7.65|2.04|**1.52**|0.776|0.974|**0.983**|
> |**Mean**|||5.15|1.48|**1.15**|0.852|0.984|**0.989**|
>
> CryoACE demonstrates superior precision and robustness in modeling complex cryo-EM protein dynamics, particularly in challenging, low-resolution cases like the CYP102A1 open state.
>
> **[W5] Computational Cost.** To ensure a fair comparison, we benchmarked all methods on a single H20 GPU using a representative ~500-residue protein system. The results are summarized below:
> |Method|Inference (min)|GPU Mem (GB)|Training|
> |---|:---:|:---:|:---:|
> |E3-CryoFold|~1|~8|~3d(8×A100)|
> |ModelAngelo|~8|~10|~5d(4×A100)|
> |CryoAtom|~10|~12|~4d(8×A100)|
> |CryoBoltz|~12|~32|N/A|
> |Boltz|~10|~24|N/A|
> |CryoACE|~15|~38|~7d(32×H20)|
>
> CryoACE's higher cost is primarily driven by the Boltz-1 foundation and self-refinement iterations, both of which our ablation confirms as essential for the reported gains. At ~15 min inference, CryoACE remains comparable to other AI methods, representing a favorable accuracy–efficiency trade-off.
>
> **[W6] Completeness Analysis.** We analyzed regions unmodeled by ModelAngelo (MA). The table below details the Fraction of MA-Unmodeled Residues and CryoACE Q-score (MA-Unmodeled) across resolutions.
> |Resolution|Fraction of MA-Unmodeled Residues|CryoACE Q-score (MA-Unmodeled)|
> |---|:---:|:---:|
> |<2.5 Å|4.5%|0.478|
> |2.5–3.5 Å |8.5%|0.330|
> |3.5–4.0 Å |17.0%|0.260|
> |Overall|10.8%|0.301|
>
> Acceptable Q-scores confirm these additionally modeled residues occupy genuine density rather than being hallucinated. MA leaves 4.5% of residues unmodeled even at high resolution (<2.5 Å), rising to 17.0% at 3.5–4.0 Å, where CryoACE still achieves a mean Q-score of 0.26 on these regions, demonstrating robust modeling where MA fails to build.
>
> **[W4/W7] Implementation Details and LaTeX Formatting.** We will provide comprehensive implementation details in the supplementary material and release our code, model weights, and dataset. All formatting issues will be fixed in the revision.

---

> > ### Author Rebuttal · Reviewer_4yu6 · 2026-04-05
> >
> > Thanks for the update and new ablations.
> >
> > [W1,2] Partially addressed. Thank you for reporting Boltz-1 and CryoBoltz baselines. One follow-up: the homogeneous test set is not sufficiently described. The manuscript mentions this is the Cryo2StructData test set, however, this is not a standard benchmark. How many structures are in the test set, and how was it constructed? It would be helpful to have it described in the paper rather than requiring readers to consult a separate paper.
> >
> > [W4] Not addressed. Implementation details are core to assessing the method, and deferring implementation details to a future manuscript and code release is insufficient for review. What is the architecture of the atom profiling module? How many self-refinement iterations k are used at inference, and how does this interact with the inference-time guidance? What are the loss hyperparameters λ1, λ2, λ3?

---

> > > ### Author Response · Authors · 2026-04-06
> > >
> > > **[W1,2]** As we mentioned in our response to Reviewer BVyF, we totally agree that this field currently lacks a standardized benchmark. To this end, we aim to ensure a fair and comprehensive evaluation across different baselines by deriving a carefully curated test set from the publicly released Cryo2StructData test split (Giri et al., 2024, 390 maps on Harvard Dataverse v1.2), which has been widely adopted by multiple recent methods including E3-CryoFold (Wang et al., 2025), CryoAtom (Su et al., 2025), and MICA (Gyawali et al., 2025). To achieve this, we 1) removed entries overlapping with the training set of the other baselines methods to avoid giving them an unfair advantage, 2) excluded RNA-containing cases to focus on protein-only evaluation, and 3) applied MMseqs2 at 30% sequence identity to filter against our own training set to prevent data leakage. This results in 276 protein chains for benchmark evaluation. We thank the reviewer for pointing out that this was not sufficiently described in the manuscript, and we will add a dedicated test set description in both the main text and supplementary material. Note that we will release all data splits and evaluation scripts upon publication.
> > >
> > > **[W4]** We thank the reviewer for pressing on this point. The key parameters are: atom profiling uses a 2-layer MLP (D=384, ReLU), k=3 self-refinement iterations, and loss weights λ1=1.0, λ2=0.5, λ3=0.1. We elaborate on each below.
> > >
> > > The atom profiling module (Eq. 5) consists of trilinear interpolation at predicted atomic coordinates followed by a learnable projection φ. The physical intuition is straightforward. In cryo-EM, the Coulomb potential peaks at atomic positions, so sampling density there yields the most informative local signal. This is also what enables iterative self-refinement, as predicted coordinates improve across iterations, the density readout at those coordinates are expected to be more accurate, producing better features for the next round. We use k=3 iterations at inference; empirically, convergence is reached by the third cycle with diminishing returns beyond that.
> > >
> > > To clarify how self-refinement interacts with guidance, each of the k=3 refinement iterations runs a full 200-step diffusion sampling process. The guidance schedule operates within each such process. In the early diffusion phase (t>0.5), the structure is still globally noisy, so we apply Sinkhorn-based global guidance (scale=1.0) to align the overall topology with the density-derived point cloud. In the late phase (t<0.3), global shape is already correct, so we switch to local Q-guidance (scale=0.5) to refine individual atomic positions against their local density fit. The gap between t=0.3 and t=0.5 is left unguided to avoid conflicting signals from the two objectives. Crucially, later refinement iterations benefit from better atom profiles, which in turn makes both global and local guidance more effective.
> > >
> > > For the loss weights (λ1=1.0, λ2=0.5, λ3=0.1), the diffusion loss dominates as the core denoising objective. The map-fitting loss is set at 0.5 rather than 1.0 because experimental density maps inevitably contain noise, and weighting it too high causes the model to fit noise artifacts rather than true structural signal. The auxiliary loss (FAPE + stereochemistry) at 0.1 serves as a soft constraint to maintain physically valid bond lengths and angles without overriding the data-driven losses. These values were selected via grid search on a held-out validation subset.
> > >
> > > We will incorporate a complete description of all architectural, training, and inference details into the revised manuscript, and release our code, model weights, and dataset upon publication.

---

### Decision · Program_Chairs · 2026-04-30

**Decision:**

Accept (regular)

**Comment:**

This paper proposes an end-to-end pipeline for challenging problem of atomic modeling from cryo-EM density maps: using Boltz as a structural prior conditioned on density maps, with inference-time guidance and iterative atom profiling to improve reconstruction accuracy.

**Rebuttal summary**
- The initial submission lacked technical details and ablation studies, raising questions about component contributions and comparison with inference-only methods. Authors substantially resolved these through additional experiments, ablations, and clarifications.
- Unresolved concern around data leakage: I think 30% sequence similarity filter used by authors is reasonable, but Boltz-1 weight initialization may introduce implicit data overlap, limiting generalizability claims.

**Recommendation: Weak Accept** Critical concerns are resolved during rebuttal. Method novelty is moderate, but strong empirical performance and the curated dataset provide additional merit. If accepted, the authors should explicitly discuss the potential influence of Boltz-1 pretrained weights on model generalizability assessment.